# N-glycosylation-defective splice variants of neuropilin-1 promote metastasis by activating endosomal signals

Xiuping Huang[1,2], Qing Ye[2,3], Min Chen[2,4], Aimin Li[1], Wenting Mi[1], Yuxin Fang[1], Yekaterina Y. Zaytseva[2,4], Kathleen L. O'Connor[2,5], Craig W. Vander Kooi[2,5], Side Liu[1] & Qing-Bai She [2,3]

Neuropilin-1 (NRP1) is an essential transmembrane receptor with a variety of cellular functions. Here, we identify two human NRP1 splice variants resulting from the skipping of exon 4 and 5, respectively, in colorectal cancer (CRC). Both NRP1 variants exhibit increased endocytosis/recycling activity and decreased levels of degradation, leading to accumulation on endosomes. This increased endocytic trafficking of the two NRP1 variants, upon HGF stimulation, is due to loss of N-glycosylation at the Asn150 or Asn261 site, respectively. Moreover, these NRP1 variants enhance interactions with the Met and β1-integrin receptors, resulting in Met/β1-integrin co-internalization and co-accumulation on endosomes. This provides persistent signals to activate the FAK/p130Cas pathway, thereby promoting CRC cell migration, invasion and metastasis. Blocking endocytosis or endosomal Met/β1-integrin/FAK signaling profoundly inhibits the oncogenic effects of both NRP1 variants. These findings reveal an important role for these NRP1 splice variants in the regulation of endocytic trafficking for cancer cell dissemination.

[1] Guangdong Provincial Key Laboratory of Gastroenterology, Department of Gastroenterology, Nanfang Hospital, Southern Medical University, Guangzhou 510515, China. [2] Markey Cancer Center, University of Kentucky College of Medicine, Lexington, KY 40506, USA. [3] Department of Pharmacology and Nutritional Sciences, University of Kentucky College of Medicine, Lexington, KY 40506, USA. [4] Department of Toxicology and Cancer Biology, University of Kentucky College of Medicine, Lexington, KY 40506, USA. [5] Department of Molecular and Cellular Biochemistry, University of Kentucky College of Medicine, Lexington, KY 40506, USA. Correspondence and requests for materials should be addressed to S.L. (email: liuside@163.com) or to Q.-B.S. (email: qing-bai.she@uky.edu)

The human neuropilin-1 (*NRP1*) gene encoding a transmembrane protein is located on chromosome 10p12 and consists of 17 exons[1,2]. NRP1 acts as a co-receptor for several growth factors including VEGF, TGF-β, HGF, FGF and PDGF, and exhibits versatile functions for neuronal axon guidance, angiogenesis and cancer initiation, growth, and metastasis[3–6]. The multifunctional capacity of NRP1 is attributed to its four ligand-binding domains (a1, a2, b1, b2), a membrane proximal MAM domain (c), and a cytoplasmic C-terminal domain[5,7]. NRP1 interacts with several receptor tyrosine kinases (VEGFR, Met, EGFR) and other transmembrane proteins (integrins, plexins/semaphorins) to elicit a range of intracellular signaling cascades initiated by specific ligands[5,7–14]. Upon ligand binding, receptors are internalized by endocytosis and transported to early and late endosomes before either recycling back to the plasma membrane or selected for degradation[15]. Endosomal signaling activated by endocytosis of receptor tyrosine kinases such as Met and EGFR play a critical role in cellular functions including, organism development and cancer progression[16–18].

Several human NRP1 variants generated by alternative splicing mechanisms have been reported[2,19–21]. NRP1 splice variants lacking the transmembrane domain are soluble proteins that bind VEGF$_{165}$, and exert anti-angiogenic and anti-tumorigenic effects by reducing NRP1 bound to growth factors and inhibiting their downstream signals[2,19,22]. In contrast, the NRP1-ΔE16 variant, which skips exon 16, does not demonstrate any functional difference as compared to the wild type (WT) NRP1[21]. NRP1 also undergoes post-translational glycosylation modifications including O-linked glycosylation at site Ser612[23] and N-linked glycosylation at several putative asparagine residues[24,25]. The O-linked glycosylation of NRP1 at Ser612 plays an important role in the modulation of VEGF signaling, cell proliferation and migration and cancer invasion[23,26,27]. However, the role of N-linked glycosylation of NRP1 is poorly characterized and understood.

In this study, we discover two human NRP1 splice variants from colorectal cancer (CRC) cell lines and tissue specimens. The two NRP1 variants are generated by skipping exon 4 and exon 5, resulting in defects in N-linked glycosylation at asparagine (N) positions N150 and N261, respectively. The altered N-linked glycosylation plays a critical role in regulation of the endocytic trafficking of NRP1 and its associated receptors, Met and β1-integrin, for CRC cell migration, invasion and dissemination. Furthermore, our work highlights that CRC expressing these NRP1 splice variants could potentially be targeted by blocking endocytosis of them and their binding receptors such as Met or β1-integrin, or their endosomal signals on activation of the FAK/p130Cas pathway.

## Results

**Identification of two NRP1 splice variants in CRC**. NRP1 acts as a signaling hub on the cell surface and plays multifaceted roles in multiple cancers including CRC[5,22]. To explore the functional importance of NRP1 in CRC progression, we cloned *NRP1* cDNA from an HCT116 CRC cell library by RT-PCR using primers at the 5′ and 3′ends of the full-length human WT *NRP1* open reading frame (2772 bp cDNA encoding 923 amino acids). Surprisingly, sequence analysis of *NRP1* cDNA clones led to characterization of two *NRP1* alternatively spliced transcripts: one that skipped exon 4 with exon 3–5 splicing (*NRP1*-ΔE4, Fig. 1a, c) and another one that skipped exon 5 with exon 4–6 splicing (*NRP1*-ΔE5, Fig. 1b, c) without a shift in the reading frame. The resulting NRP1-ΔE4 lacks 76 amino acids (position 144–219 in the front region of the a2 domain) encoded by exon 4, and NRP1-ΔE5 lacks 52 amino acids (position 220–271 in the back region of the a2 domain) encoded by exon 5 (Fig. 1c; Supplementary

Fig. 1). Interestingly, the exon-intron junctions of the *NRP1* gene contain the alterative splice consensus (5′GT/AG3′) sequence[1] for generation of the *NRP1*-ΔE4 and *NRP1*-ΔE5 variants. To further validate the existence of these variants, RT-PCR of the region that includes exons 4 and 5 was performed with several human CRC cell lines and a subset of CRC tissues as well as the adjacent nonmalignant colonic mucosa using the forward primer in exon 3 and the reverse primer in exon 6. Three bands were visualized on gel electrophoresis in HCT116 cells: the expected *NRP1*-WT 532-bp, along with *NRP1*-ΔE4 376-bp and *NRP1*-Δ5 304-bp fragments (Fig. 1d). Notably, the abnormal *NRP1* DNA fragments were found in a subset of primary colorectal tumors (NF99, NF103, NF105, NF106, NF110) and the primary CRC and liver metastasis tumor cell lines (Pt93, Pt2377 and LM2377) at an expression level equal to or above that found for *NRP1*-WT mRNA (Fig. 1e, f; Supplementary Table 1). Of these, two CRC tissues (NF99, NF106) expressed the variants as dominant NRP1 isoforms (Fig. 1e, f). In contrast, the abnormal *NRP1* DNA fragments were not detected in non-malignant colonic mucosa even when using increased amounts of RNA for RT-PCR analysis, whereas *NRP1*-WT could be detected in these normal tissues by increasing their RNA levels (Fig. 1e; Supplementary Fig. 2a). Direct sequencing of the RT-PCR products confirmed that the three bands correspond to *NRP1*-WT, *NRP1*-ΔE4 and *NRP1*-ΔE5, respectively. Additional analysis of tumor samples from 126 patients with stage I–IV CRC showed that NRP1-ΔE4 was positively expressed in 71% of CRC tissues and significantly enriched with CRC progression, whereas NRP1-ΔE5 expression was expressed less frequently (25%) and not significantly observed as tumors progressed through stages I–IV (Table 1; Supplementary Table 2). Together, these findings reveal two human NRP1 splice variants that have never been investigated, of which the NRP1-ΔE4 expression correlates with CRC progression.

**The NRP1 variants accumulate in intracellular compartments**. While the mRNA expression of WT *NRP1* and its two splice variants was detected in the HCT116 CRC cell line, their protein expression was barely detected in this cell line (Supplementary Fig. 2b). To characterize the specific function of the two NRP1 variants, the NRP1-WT, ΔE4 and ΔE5 were stably expressed at comparable levels in HCT116 and HT29 CRC cells (Fig. 2a). Expression of NRP1-WT, ΔE4 or ΔE5 in these cells did not affect the protein levels of EGFR and Met receptors (Fig. 2a), whereas VEGFR2 expression was not found in these two cell lines. NRP1-WT was expressed at the plasma membrane in both cell lines with regular culture conditions containing 10% fetal bovine serum (FBS) as evidenced using the membrane protein α6-integrin as a positive control (Fig. 2b, c). Interestingly, NRP1-ΔE4 and NRP1-ΔE5 were expressed predominantly in punctate cytoplasmic structures (Fig. 2b, c). Similar results were observed at 48 h after transient transfection with the NRP1-WT and its two variants in HCT116 and HT29 CRC cells, although the two NRP1 variants initially localized at the plasma membrane at 24 h after the transfection (Supplementary Fig. 2c, d). Notably, expression of endogenous NRP1-ΔE4 and NRP1-ΔE5 proteins and their intracellular accumulation could also be detected in the Pt93, Pt2377 and LM2377 primary CRC cell lines; of these Pt93 cells expressed both NRP1-ΔE4 and NRP1-ΔE5 proteins (Supplementary Fig. 2e, f). Separation procedures using a cell surface biotinylation assay demonstrated that ~70% of the two variants was present in the intracellular fraction compared with 5% of the NRP1-WT (Fig. 2d, e). NRP1 is well-known as a co-receptor for VEGF$_{165}$[6]. In the absence of ligand, NRP1-WT, NRP1-ΔE4 and NRP1-ΔE5 were predominantly distributed at the plasma membrane (Fig. 2f). VEGF$_{165}$ largely induced the full internalization of

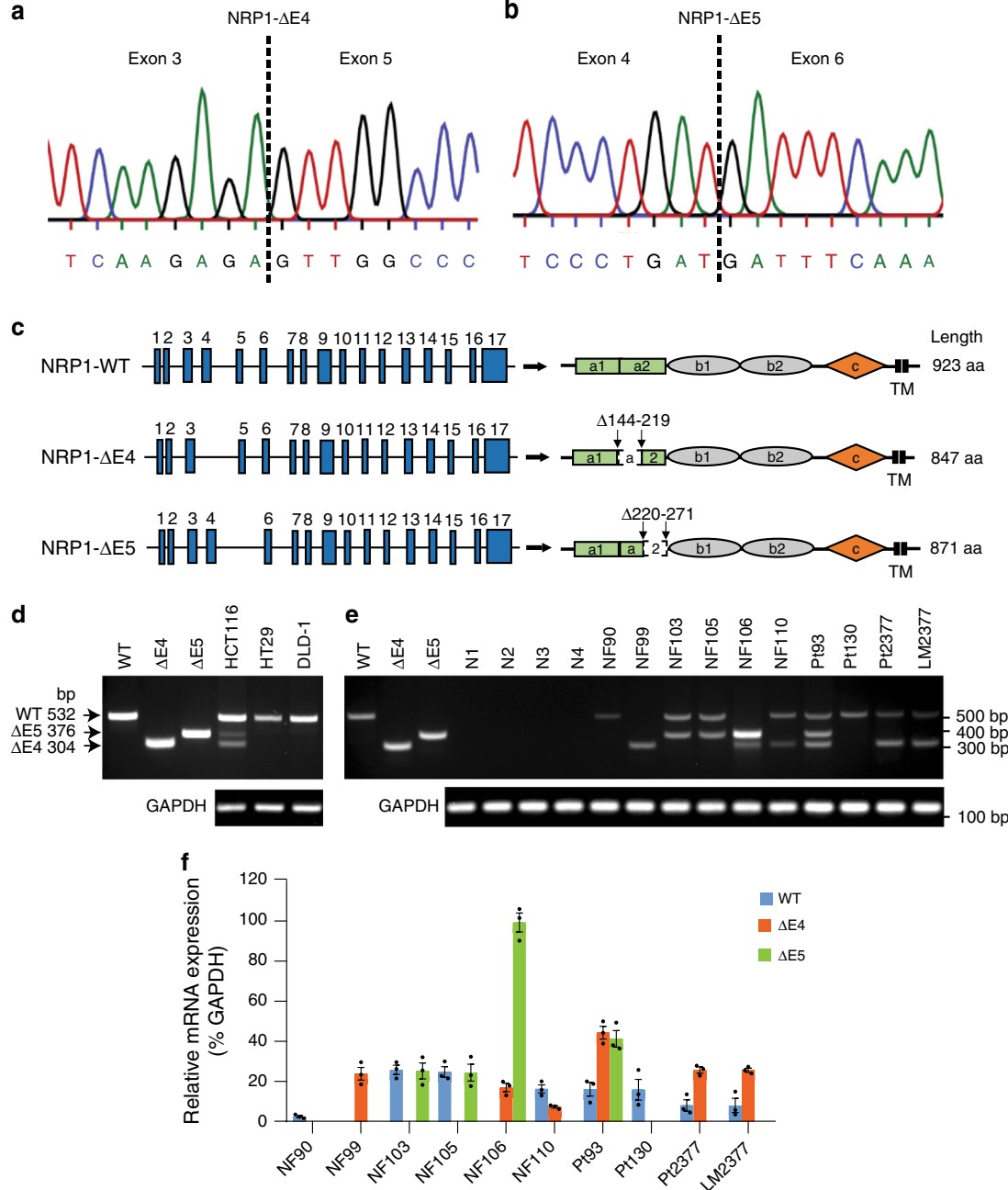

**Fig. 1** NRP1-ΔE4 and NRP1-ΔE5 are commonly expressed in CRC. **a**, **b** Sequencing analysis of several human *NRP1* cDNA clones identified two NRP1 splice variants: a splicing of exon 3 to exon 5 **a** and a splicing of exon 4 to exon 6 **b**. **c** Schematics of genomic (left) and protein (right) structures of the *NRP1* gene and the full-length WT NRP1 as well as two identified splice variants, NRP1-ΔE4 and NRP1-ΔE5. **d**, **e** RT-PCR analysis was performed on total RNA isolated from CRC cell lines **d**, and normal (N1-4) and CRC tissues, as well as primary CRC cell lines **e**. The high, middle and low molecular weight products of 532 bp, 376 bp and 304 bp are amplified from NRP1-WT, NRP1-ΔE5 and NRP1-ΔE4, respectively. GAPDH amplification was used as control for RT-PCR. **f** The expression levels of NRP1-WT, NRP1-ΔE4 and NRP1-ΔE5 relative to GAPDH expression levels in CRC tissues as shown in **e** are quantified using Image J software. The data are presented as mean ± s.e.m. (*n* = 3 independent experiments)

NRP1-WT from the plasma membrane to the perinuclear region, but the two NRP1 variants showed little or no internalization upon VEGF$_{165}$ or EGF stimulation (Fig. 2f–h). In contrast, HGF, another NRP1-binding ligand[28,29] that is often overexpressed in CRC[30], markedly induced endocytosis of NRP1-ΔE4 and NRP1-ΔE5, but had modest effect on the internalization of NRP1-WT (Fig. 2f, i). Depletion of HGF from FBS using an anti-HGF neutralizing antibody dramatically reduced the internalization of the two NRP1 variants (Supplementary Fig. 2g, h). Collectively, these data indicate that HGF is a critical ligand to induce

internalization of the two NRP1 splice variants and their accumulation in intracellular compartments.

**The NRP1 variants shuttle between cell surface and endosomes.** We next determined the NRP1-WT, ΔE4, and ΔE5-containing vesicular compartments by staining for several intracellular vesicle trafficking markers. Immunofluorescence analysis revealed a significantly high level of co-localization between the NRP1 splice variants and early endosomal antigen 1 (EEA1) compared with NRP1-WT in HT29 cells (Fig. 3a, c). NRP1-ΔE4

**Table 1 Expression of NRP1-ΔE4 and NRP1-ΔE5 in CRC**

|  | NRP1-ΔE4 | | | NRP1-ΔE5 | | |
|---|---|---|---|---|---|---|
|  | **Negative** | **Positive** | **Total** | **Negative** | **Positive** | **Total** |
| Stage I | 18 | 12 | 30 | 27 | 3 | 30 |
| Stage II | 11 | 36 | 47 | 34 | 13 | 47 |
| Stage III | 3 | 26 | 29 | 20 | 9 | 29 |
| Stage IV | 4 | 16 | 20 | 13 | 7 | 20 |
| Total | 36 | 90 | 126 | 94 | 32 | 126 |
|  | $p = 0.00013$ | | | $p = 0.15$ | | |

Statistical significance of NRP1-ΔE4 or NRP1-ΔE5 expression toward CRC progression was determined by the $\chi^2$-test with $p$-value indicated

and NRP1-ΔE5 accumulated in the perinuclear area and co-localized more with the late endosomal marker Rab7 and three well-known recycling endosomal markers: transferrin, Rab4 and Rab11 (Fig. 3b, c; Supplementary Fig. 3a). No significant co-localization was detected with lysotracker or lysosomal cathepsin D protease (Supplementary Fig. 3b, c). Similar results were observed in HCT116 cells expressing NRP1-WT and its two variants (Supplementary Fig. 3d–g). Additionally, intracellular accumulation of endogenous NRP1-ΔE4 and NRP1-ΔE5 in the primary CRC cell lines (Pt93, Pt2377 and LM2377) was also found in endosomes as demonstrated by co-localization with Rab7 (Supplementary Figs. 2e, 3h). Thus, under basal conditions, NRP1-ΔE4 and NRP1-ΔE5 localized mostly in early, late and recycling endosomes and less on the membrane surface (Fig. 2d). These data suggest that NRP1-ΔE4 and NRP1-ΔE5 constitutively

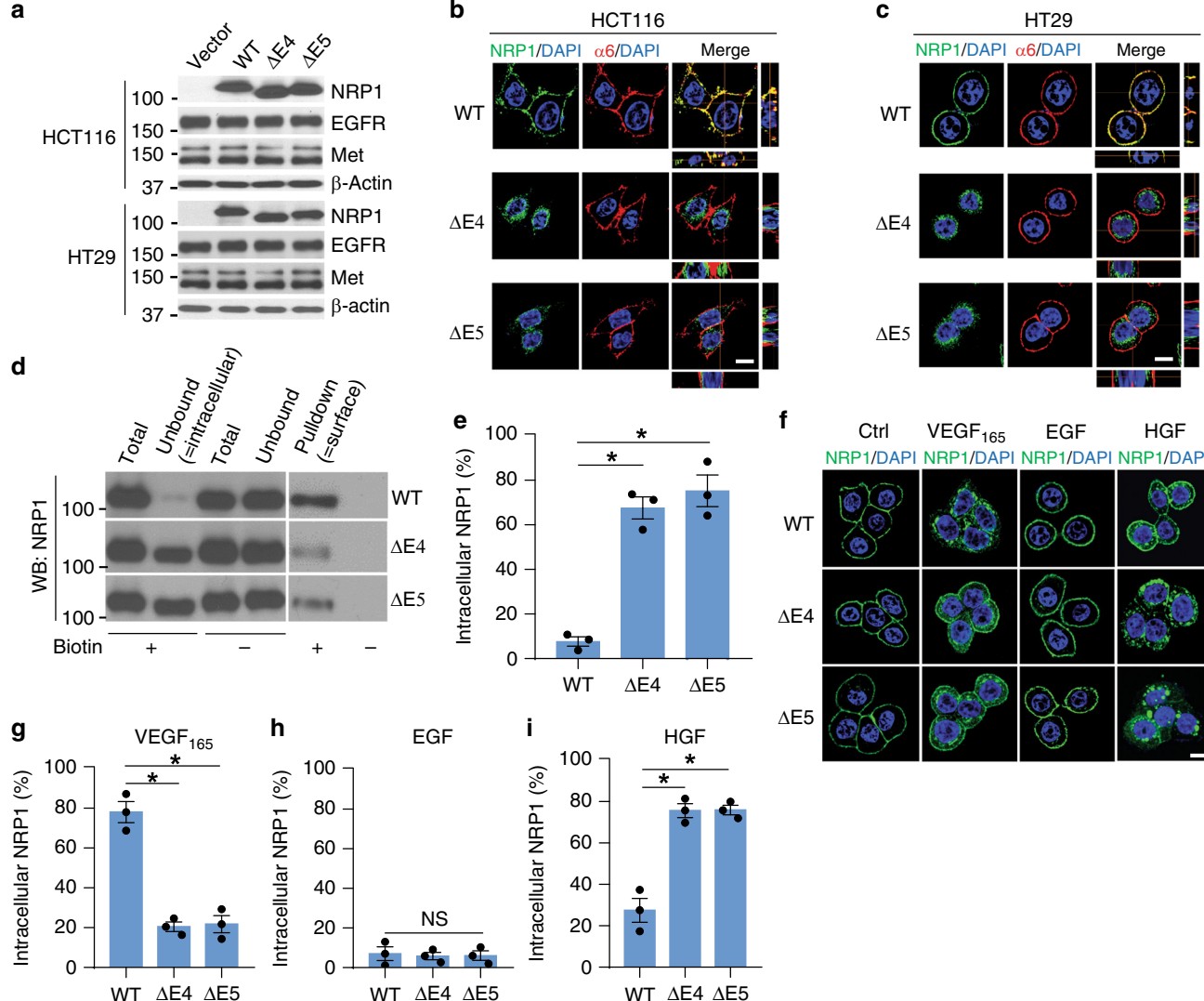

**Fig. 2** NRP1-ΔE4 and NRP1-ΔE5 accumulate in intercellular compartments. **a** HCT116 and HT29 cells with stable expression of NRP1-WT, NRP1-ΔE4, NRP1-ΔE5 or vector control were assessed by western blot analysis. **b, c** Confocal images and z-stack projections of transverse sections of HCT116 **b** and HT29 **c** cells expressing the indicated NRP1 isoforms, stained for NRP1, α6-integrin and DAPI. Scale bars, 10 μm. **d** HT29 cells with expression of the indicated NRP1 isoforms were surface-biotinylated, then the biotinylated proteins were pulled down by streptavidin beads. The total sample before pulldown (total), the supernatant corresponding to the intracellular fraction (unbound) and the surface fractions (bound) were analyzed by western blot for NRP1. **e** The percentage of intracellular NRP1-WT and the variants was calculated as a ratio of the total. **f** Confocal images of NRP1 with DAPI staining in serum-starved HT29 cells with expression of the indicated NRP1 isoforms, stimulated with VEFG$_{165}$ (50 ngml$^{-1}$), EGF (50 ngml$^{-1}$), HGF (50 ngml$^{-1}$) or PBS as control for 30 min. Scale bars, 10 μm. **g–i** The percentage of intracellular NRP1 staining upon stimulation with VEFG$_{165}$, EGF or HGF as shown in **f**. All graphic data are presented as mean ± s.e.m. ($n = 3$ independent experiments). *$p < 0.002$; NS not significant using Student's $t$-test

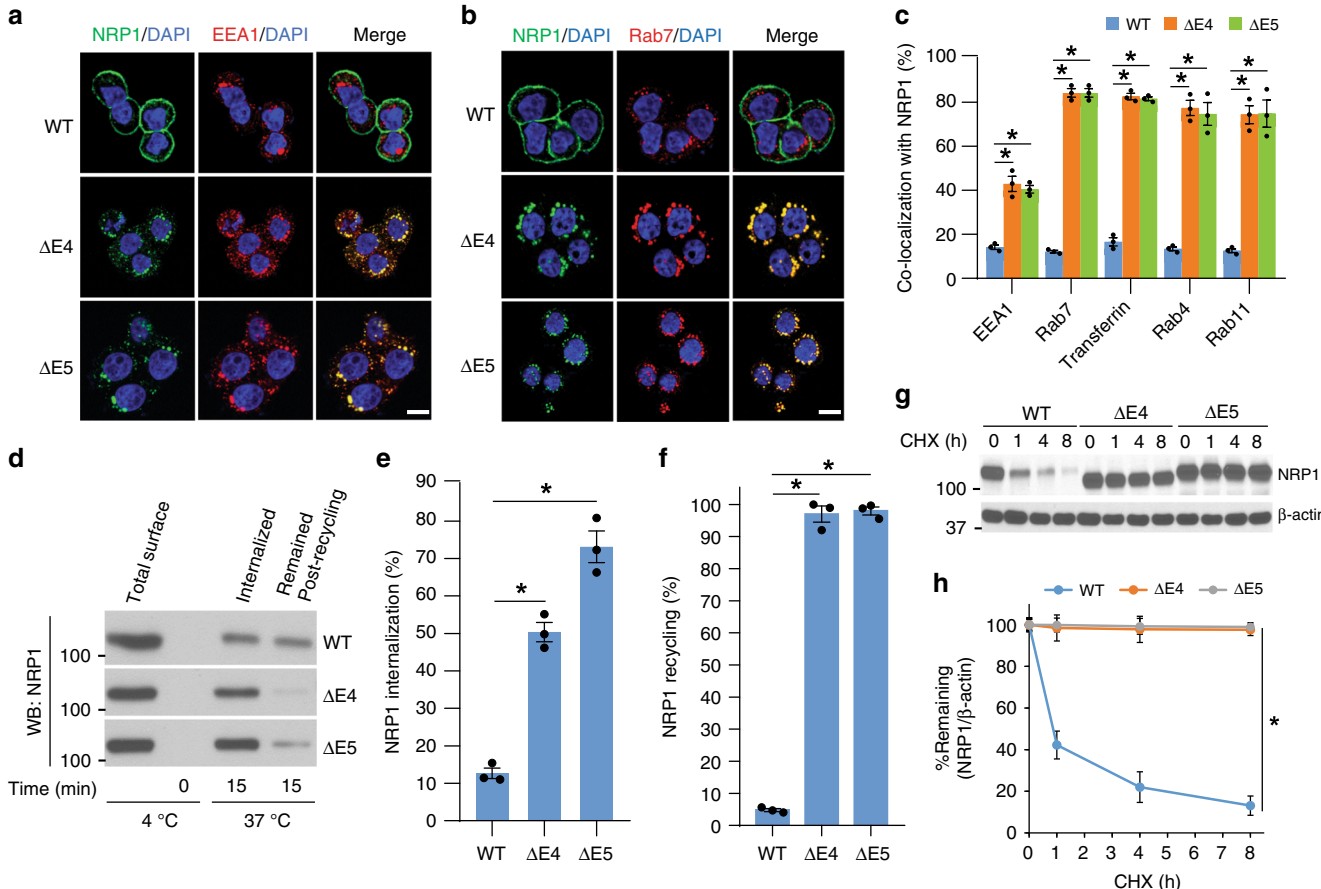

**Fig. 3** The NRP1 splice variants shuttle between cell surface and endosomes and escape from degradation. **a, b** Confocal images of NRP1, EEA1, Rab7 and DAPI staining in HT29 cells with expression of the indicated NRP1 isoforms. Scale bars, 10 μm. **c** HT29 cells with expression of the indicated NRP1 isoforms were incubated with Texas-Red-transferrin for 30 min before fixation, and then stained for NRP1, EEA1, Rab7, Rab4 or Rab11 and DAPI (Fig. 3a, b and Supplementary Fig. 3a). The percentage of co-localization between NRP1-WT or the splice variants and endosomal markers EEA1, Rab7, transferrin, Rab4 or Rab11 was analyzed using Nikon NIS-Elements AR software. **d-f** Biotinylaion internalization/recycling analysis. In HT29 cells with expression of the indicated NRP1 isoforms, the levels of surface biotinylated NRP1-WT or the splice variants that were internalized (15 min) or recycled (15 min) were measured by western blot for NRP1 after streptavidin pulldown **d**. Quantification of internalization **e** and recycling **f** of NRP1-WT and the splice variants through densitometric analysis of bands from western blots **d** using Image J software. **g** HCT116 cells with stable expression of NRP1-WT, NRP1-ΔE4 or NRP1-ΔE5 were treated with 50 μgml$^{-1}$ cycloheximide (CHX) for the indicated times followed by western blot analysis. **h** The western blots of NRP1 shown in **g** were quantified using Image J software. The level of NRP1 remaining was obtained by normalizing to the β-actin level at each time point. All graphic data are presented as mean ± s.e.m. ($n = 3$ independent experiments). $*p < 0.001$ using Student's $t$-test

internalize and recycle back to the cell surface or may secrete to the extracellular space.

To determine the rates of NRP1 internalization and subsequent recycling, surface-biotinylated NRP1 isoforms were measured by streptavidin-agarose pulldown after different incubation periods. After 15 min incubation with regular culture medium, 50–73% of the two NRP1 splice variants were internalized versus 12% of the NRP1-WT (Fig. 3d, e). On re-incubation at 37 °C for an additional 15 min, the remaining internalized NRP1 splice variants were much lower when compared with NRP1-WT (Fig. 3d), indicating that the NRP1 splice variants exhibit an increase in their recycling activities (Fig. 3f). Notably, cycloheximide chase analysis for protein degradation showed that NRP1-ΔE4 and NRP1-ΔE5 were not significantly degraded between 0 and 8 h with regular culture medium or with HGF stimulation as compared to the rapid degradation of NRP1-WT (Fig. 3g, h; Supplementary Fig. 3i, j). Thus, the two NRP1 splice variants accumulate on endosomal compartments through constitutive shuttling between the cell surface and endosomes and display defective degradation.

**Defect in N-linked glycosylation increases NRP1 endocytosis.** Within the deleted regions of NRP1-ΔE4 or NRP1-ΔE5, we found putative N-linked glycosylation sites at asparagine residues N150 and N261, respectively[24,25] with the consensus motif, N-X-S/T[31] (Supplementary Fig. 1). To determine whether N150 and N261 are specific N-linked glycosylation sites that affect endocytic trafficking of NRP1-ΔE4 and NRP1-ΔE5, respectively, NRP1 mutants in which N150 and/or N261 sites were replaced with glutamine (Q) were expressed in HCT116 cells. Cell lysates were treated with PNGase F, a glycosidase enzyme that hydrolyzes N-linked oligosaccharides, and then analyzed by a migration shift assay. We found that NRP1-WT, ΔE4, ΔE5, and N150Q and N261Q mutants shifted downward after enzyme treatment (Fig. 4a, b). However, the double mutant N150Q/N261Q displayed a lower molecular weight similar to that of the glycosidase-treated NRP1-WT (Fig. 4a, b). These data suggest that the N-linked glycans were added on both N150 and N261 sites of NRP1. Similar to the NRP1-ΔE4 and NRP1-ΔE5 variants, both NRP1 single mutants N150Q and N261Q that were stably or transiently (48 h) expressed in HCT116 or HT29 cells also

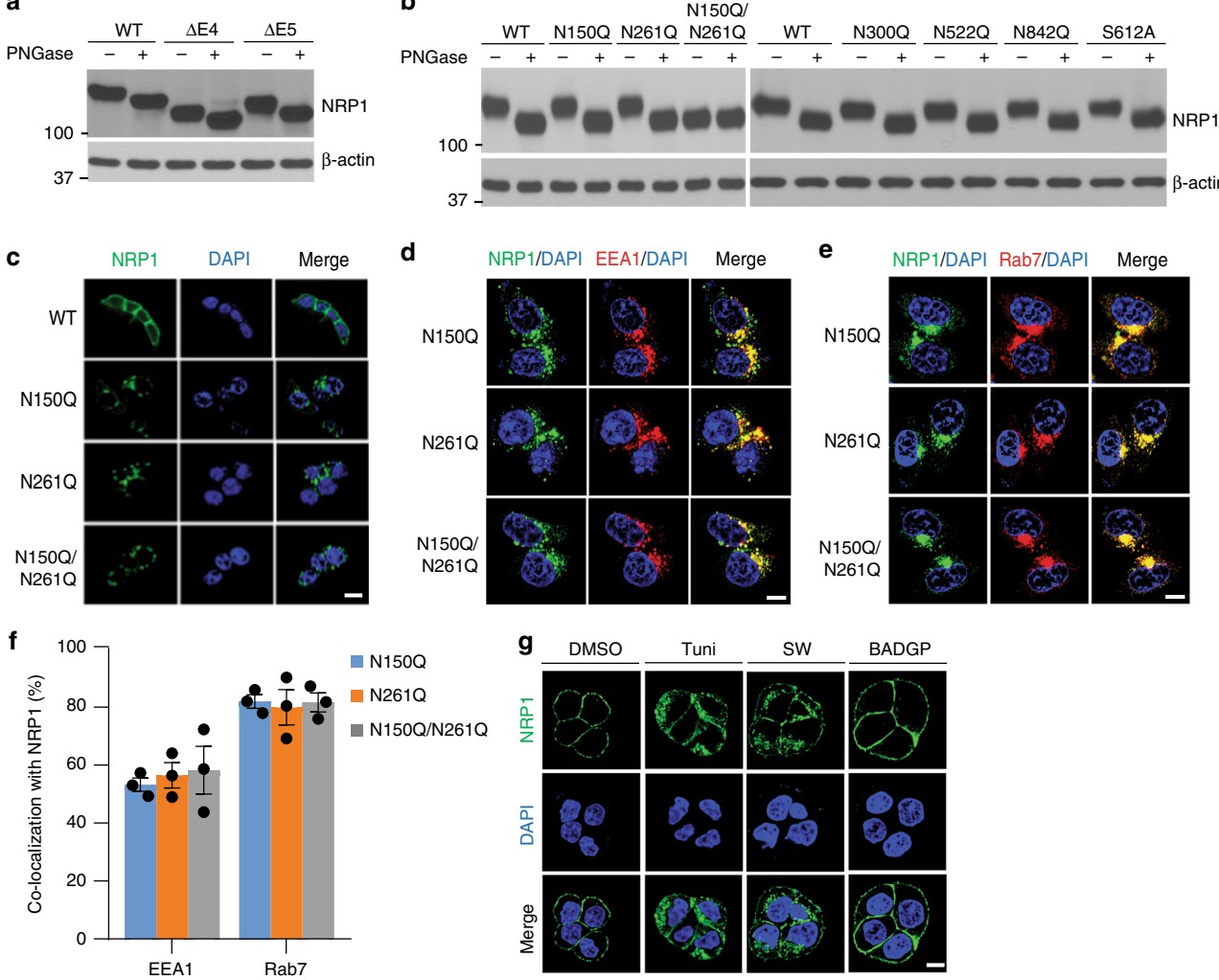

**Fig. 4** Defect in N-glycosylation at N150 or N261 in NRP1 leads to increased endocytic trafficking. **a**, **b** The lysates of HCT116 cells with stable **a** or transient **b** expression of NRP1-WT or the indicated NRP1 splice variants or mutants were treated with or without PNGase, followed by western blot analysis. **c** Confocal images of NRP1 with DAPI staining in HCT116 with stable expression of NRP1-WT or its mutants. Scale bars, 10 μm. **d**, **e** Confocal images of NRP1, EEA1, Rab7 and DAPI staining in HCT116 cells with stable expression of NRP1-WT or its mutants. Scale bars, 10 μm. **f** The percentage of co-localization between NRP1 and EEA1 or Rab7 as shown in **d** and **e** is presented as mean ± s.e.m. ($n = 3$ independent experiments). **g** Confocal images of NRP1 with DAPI staining in HT29 cells with stable expression of NRP1-WT that were treated with tunicamycin (10 μgml⁻¹), swainsonine (5 μgml⁻¹), BADGP (2 mM) or DMSO as control for 24 h. Scale bars, 10 μm

accumulated in the perinuclear area and co-localized with endosome markers EEA1 and Rab7 with no significant degradation (Fig. 4c–f; Supplementary Fig. 4a–d). Interestingly, these findings were obtained similarly with the double mutant N150Q/N261Q (Fig. 4c–f; Supplementary Fig. 4a-d). By contrast, three other putative N-glycosylation site mutants, N300Q, N522Q and N842Q[25] (Supplementary Fig. 1) and the O-linked glycosylation-deficient mutant S612A[23] shifted downward after treatment with PNGase F (Fig. 4b). These mutants were expressed on the cell surface with rapid degradation after exposure to cycloheximide similar to NRP1-WT (Supplementary Fig. 4b–d). Furthermore, exposure of cells to inhibitors of N-glycan synthesis (tunicamycin or swainsonine) largely induced NRP1 accumulation in intracellular compartments, while an inhibitor of O-glycosylation (BADGP) did not (Fig. 4g). In addition, HGF stimulation, while not VEGF₁₆₅, completely induced the endocytosis of both N150Q and N261Q NRP1 mutants, but NRP1-WT and its mutants N300Q, N522Q, N842Q and S612A were internalized to a lesser extent with HGF stimulation (Supplementary Fig. 4e). Collectively, these data demonstrate that N150 and N261-linked

glycosylation modification are critical in regulation of NRP1 endocytic trafficking in response to HGF stimulation.

**N-glycosylation-defective NRP1 promotes CRC metastasis.** NRP1 is involved in regulation of a variety of cellular functions, such as cell proliferation, migration and invasion associated with cancer progression[5]. Analysis of cell growth rate showed no significant difference with stable expression of NRP1-WT, ΔE4 or ΔE5 when compared to vector control in HCT116 cells (Supplementary Fig. 5a). Expression of NRP1-WT in HCT116 cells significantly enhanced cell migration and invasion as determined by Boyden chamber assays and by tracking single-cell movement with live cell imaging under basal conditions (Fig. 5a–c; Supplementary Fig. 5b, c). However, the two NRP1 splice variants showed enhanced cell migratory and invasive capabilities (2–3 fold) when compared with NRP1-WT (Fig. 5a–c; Supplementary Fig. 5b, c). HGF stimulation markedly enhanced cell migration in HCT116 cells expressing NRP1-ΔE4 or NRP1-ΔE5 compared with cells expressing NRP1-WT, whereas these effects were not

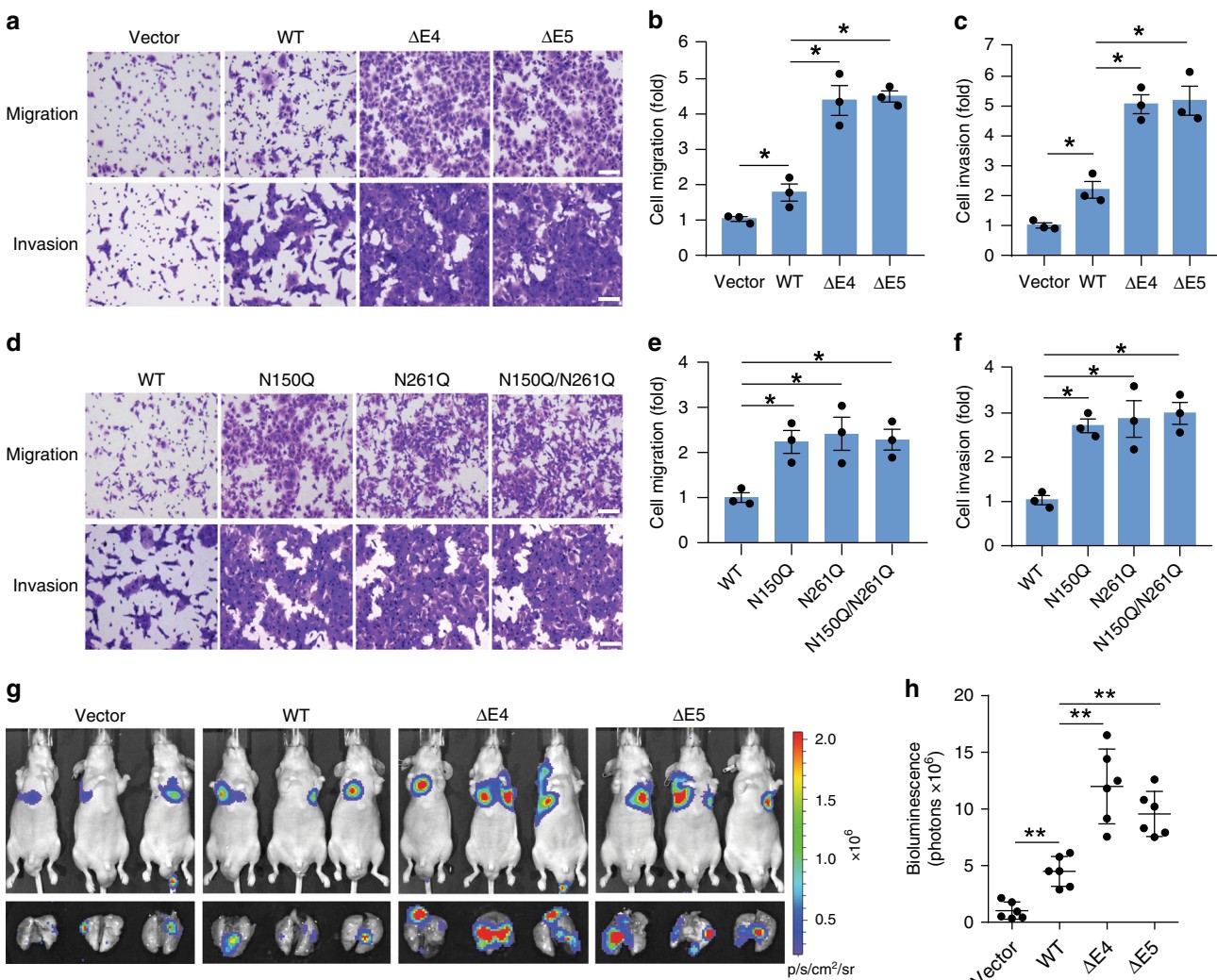

**Fig. 5** N-glycosylation-defective NRP1 splice variants increases metastatic capacity. **a–c** Transwell migration and invasion analyses of HCT116 cells with stable expression of NRP1-WT, NRP1-ΔE4, NRP1-ΔE5 or vector control over 6 h and 30 h of incubation, respectively. Scale bars, 100 μm. The number of migrated and invaded cells per field ($n = 5$) were counted. The results were expressed as a fold increase over the migrated **b** or invaded **c** cell number found in vector control cells. **d–f** Transwell migration and invasion analyses of HCT116 cells with stable expression of NRP1-WT or mutants over 6 h and 30 h of incubation, respectively. Scale bars, 100 μm. The number of migrated cells per field ($n = 5$) were counted. The results are expressed as a fold increase over the migrated **e** or invaded **f** cell number found in NRP1-WT cells. **g** Bioluminescence images of lung metastasis in athymic nude mice that were injected intravenously with HCT116-Luciferase/GFP cells expressing vector, NRP1-WT, NRP1-ΔE4 or NRP1-ΔE5 at week 6 post-injection. **h** Quantitative analysis of bioluminescence in lung metastasis as shown in **g** was performed, and the results are presented as mean ± s.e.m. ($n = 6$ mice/group). The graphic data in **b**, **c**, **e**, **f** are presented as mean ± s.e.m. ($n = 3$ independent experiments). *$p < 0.04$ using Student's $t$-test; **$p < 0.005$ using Mann–Whitney test

observed with VEGF$_{165}$ stimulation (Supplementary Fig. 5d). Conversely, silencing NRP1 in Pt93 and LM2377 primary CRC cells that express higher levels of endogenous NRP1-ΔE4 and/or NRP1-ΔE5 than NRP1-WT (Supplementary Fig. 2e) resulted in marked inhibition of cell migration and invasion (Supplementary Fig. 5e-g). Similar to the NRP1-ΔE4 and NRP1-ΔE5 variants, both N-glycosylation-defective NRP1 mutants N150Q and N261Q profoundly enhanced cell migration and invasion compared to NRP1-WT (Fig. 5d–f). By contrast, the double mutant N150Q/N261Q showed no enhancement in these capabilities when compared with either of the single mutants (Fig. 5d–f). To determine whether the NRP1 variants facilitate cancer metastasis, we used an experimental lung metastasis model in vivo. Luciferase-labeled HCT116 cells with stable expression of NRP1-WT, NRP1-ΔE4, NRP1-ΔE5 or vector control were injected intravenously into athymic nude mice, and lung metastasis was

assessed by bioluminescent imaging. Compared to NRP1-WT and vector control, expression of either of the NRP1 splice variants remarkably promoted lung metastases in mice (Fig. 5g, h). Thus, these findings reveal an important role of the NRP1 splice variants in CRC cell dissemination.

**Met and β1-integrin co-internalize with the NRP1 variants.** NRP1 interacts with the tyrosine kinase receptor Met and β1-integrin to increase malignant transformation activity[10,12,13]. Considering that both Met and β1-integrin play important roles in CRC progression and metastasis[30,32,33], we examined the possible difference between NRP1-WT and its splice variants in the interaction and co-localization with Met and β1-integrin in CRC cells. Co-immunoprecipitation analysis showed that, compared with NRP1-WT, the two NRP1 variants had greater

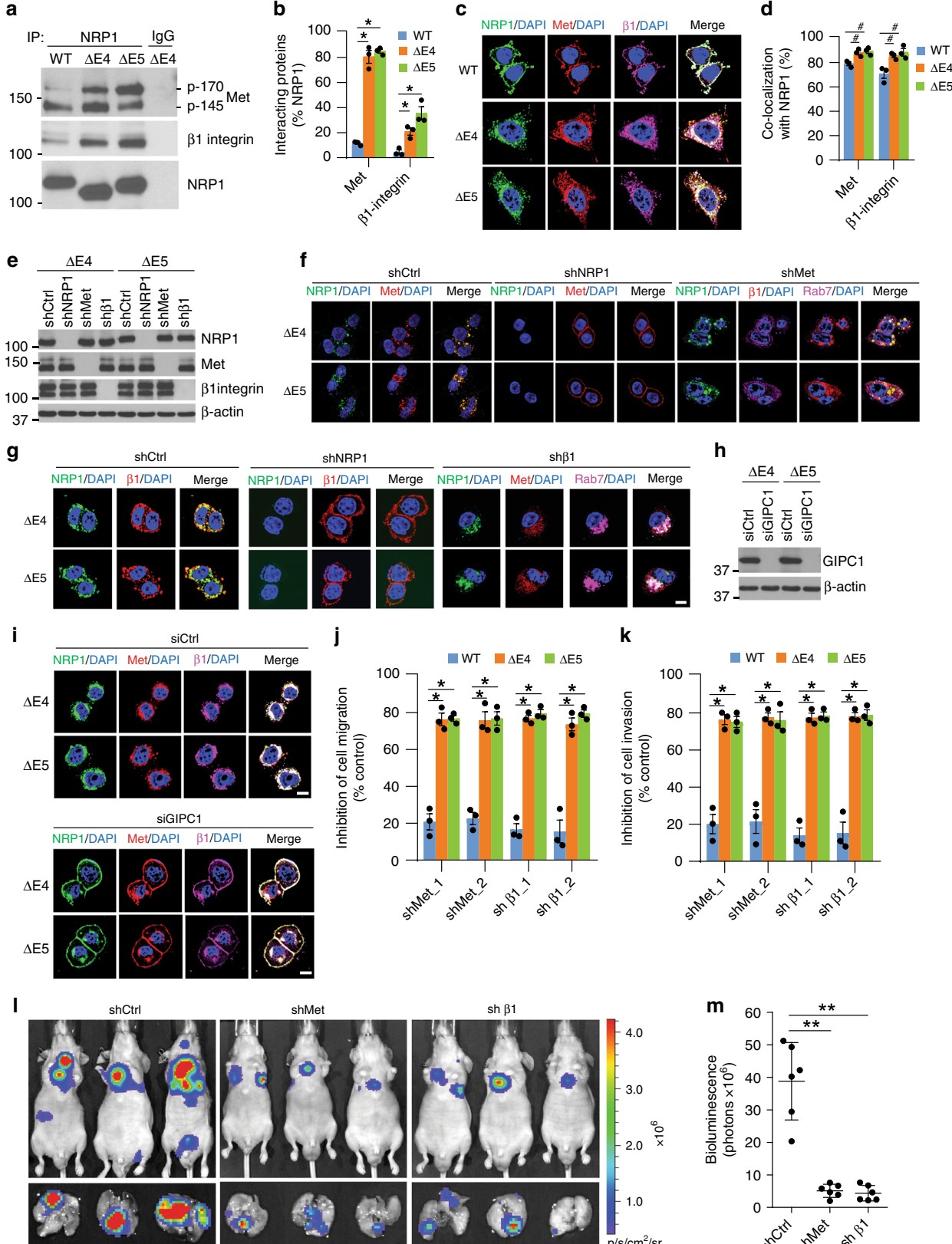

interaction with both the precursor (p-170) and mature β-chain (p-145) forms of Met, as well as with β1-integrin (Fig. 6a, b). Remarkably, immunofluorescence staining and separation procedures using a cell surface biotinylation assay revealed that Met was highly internalized and co-localized with the two NRP1 variants in intracellular compartments (Fig. 6c, d; Supplementary Fig. 6a, b). By contrast, Met co-localized with NRP1-WT on the plasma membrane (Fig. 6c, d). Interestingly, active β1-integrin (12G10 antibody)[34] was also visible predominantly in punctate cytoplasmic structures and largely

**Fig. 6** Met and β1-integrin co-internalize with the NRP1 variants for CRC cell dissemination. **a** HT29 cells with stable expression of NRP1-WT, NRP1-ΔE4 or NRP1-ΔE5 were lysed and immunoprecipitated with NRP1 antibody or IgG as control followed by western blot analysis. **b** Quantification of Met or β1-intergrin interaction with NRP1-WT and the splice variants through densitometric analysis of bands from western blots **a** using Image J software. **c** Confocal images of NRP1, Met, active β1-integrin and DAPI staining in HCT116 cells with expression of the indicated NRP1 isoforms. Scale bars, 10 μm. **d** Quantification of co-localization between Met or β1-integrin and the indicated NRP1 isoforms as shown in **c**. **e–g** NRP1-ΔE4- or NRP1-ΔE5-expressing HCT116 cells with stable expression of NRP1 shRNA, Met shRNA, β1-integrin shRNA or control shRNA were assessed by western blot analysis **e** or by confocal sections of the cells stained for NRP1, Met, β1-integrin, Rab7 and with DAPI **f**, **g**. Scale bars, 10 μm. **h**, **i** NRP1-ΔE4- or NRP1-ΔE5-expressing HT29 cells with transient expression of GIPC1 siRNA or control siRNA were assessed by western blot analysis **h** or by confocal sections of the cells stained for NRP1, Met, β1-integrin and DAPI **i**. Scale bars, 10 μm. **j**, **k** NRP1-WT-, NRP1-ΔE4- or NRP1-ΔE5-expressing HCT116 cells with stable expression of two different sets of Met shRNA, β1-integrin shRNA or control shRNA were assessed by transwell migration **j** and invasion **k** analyses over 6 h and 30 h of incubation, respectively. The results are expressed as the inhibition of cell migration or invasion relative to each of the shCtrl-expressing cells. **l** Bioluminescence images of lung metastasis in athymic nude mice that were injected intravenously with NRP1-ΔE4-expressing HCT116-Luciferase/GFP cells with stable expression of Met shRNA, β1-integrin shRNA or control shRNA at week 6 post-injection. **m** Quantitative analysis of bioluminescence in lung metastasis as shown in **l** was performed, and the results are presented as mean ± s.e.m. ($n = 6$ mice/group). The graphic data in **b**, **d**, **j**, **k** are presented as mean ± s.e.m. ($n = 3$ independent experiments). *$p < 0.005$; #$p < 0.03$ using Student's $t$-test; **$p < 0.005$ using Mann–Whitney test

co-localized with both NRP1 variants and Met in the perinuclear area (Fig. 6c, d). However, active β1-integrin co-localized with NRP1-WT and Met on the plasma membrane in NRP1-WT-expressing cells under basal conditions (Fig. 6c, d). Similar results were observed in primary CRC cell lines (Pt93, Pt2377, LM2377) that express endogenous NRP1-ΔE4 and/or -ΔE5 (Supplementary Figs. 2e, 6c, d, 7a) and in HT29 cells expressing the N-glycosylation-defective NRP1 mutant N150Q or N261Q (Supplementary Figs. 6e, 7b). Upon HGF but not $VEGF_{165}$ stimulation, Met also internalized, accumulated in the perinuclear area, and co-localized with the two NRP1 variants and NRP1 mutants N150Q and N261Q (Supplementary Fig. 6f). Moreover, Met protein levels were stable in HCT116 cells expressing either the NRP1 variants or mutants as compared to its rapid degradation in the NRP1-WT-expressing cells after treatment with cycloheximide (Supplementary Fig. 6g, h); whereas β1-integrin levels were relatively stable in the cells expressing either NRP1-WT or its variants or mutants (Supplementary Fig. 7c, d). To determine whether the internalization of Met and β1-integrin is dependent on binding to the NRP1 variants, NRP1-ΔE4 and NRP1-ΔE5 were knocked down transiently or stably by RNAi. Silencing of either NRP1-ΔE4 or NRP1-ΔE5 expression in HCT116 and HT29 cells completely prevented the internalization of Met and β1-integrin, but had no effect on Met/β1-integrin interaction (Fig. 6e–g; Supplementary Figs. 6i–k). Notably, knockdown of either Met or β1-integrin expression did not alter the accumulation of NRP1-ΔE4 or NRP1-ΔE5 on endosomes or protein stability 8 h after cycloheximide exposure (Fig. 6e–g; Supplementary Figs. 6l, m, 7e, f). However, knockdown of Met caused β1-integrin dissociation from the endosomal NRP1 variants, whereas knockdown of β1-integrin had no effect on Met interaction and co-localization with the NRP1 variants on endosomes (Fig. 6e–g; Supplementary Fig. 7g). GIPC1 is a well-known NRP1 interacting protein via direct binding to the SEA motif in the intracellular domain of NRP1[35], and couples NRP1 to other signaling receptors, including VEGFR and integrins, to accelerate their endocytosis[13,36]. Silencing GIPC1 largely abrogated the internalization of the two NRP1 variants as well as the co-internalization of Met and β1-integrin in the NRP1 variant-expressing cells under basal conditions (Fig. 6h, i). Taken together, these data indicate that internalization of NRP1-ΔE4 and NRP1-ΔE5 and their accumulation on endosomes are independent of Met or β1-integrin activity. Conversely, internalization of Met and β1-integrin and their accumulation on endosomes are controlled by the NRP1 variants and the adaptor protein GIPC1.

Notably, silencing of either Met or β1-integrin profoundly inhibited cell migration and invasion (~80%) in HCT116 cells expressing NRP1-ΔE4 or NRP1-ΔE5, compared to the modest inhibition (~20%) observed in the cells expressing NRP1-WT (Fig. 6j, k). Given that endosomal Met-mediated Rac1 activation has been shown to enhance cell migration[17], we examined whether the NRP1 variants that lead to Met internalization on endosomes also activate Rac1. We found that Rac1 was substantially activated in HCT116 cells expressing NRP1-ΔE4 or NRP1-ΔE5 in regular growth medium or upon HGF stimulation (Supplementary Fig. 8a, b), and silencing Rac1 profoundly inhibited cell migration in these cells (Supplementary Fig. 8c, d). Furthermore, lung metastasis induced by NRP1-ΔE4 in vivo was also dramatically repressed by knockdown of either Met or β1-integrin expression (Fig. 6l, m). These findings highlight that both Met and β1-integrin function as key partners of the NRP1 variants in endosomes to promote cell migration, invasion and metastasis.

**Endocytosis is required for the NRP1 variants' activities.** Clathrin-dependent and Clathrin-independent endocytosis require the activity of dynamin, a GTPase responsible for pinching vesicles from the plasma membrane and thereby driving internalized cargo into carrier vesicles[15]. To determine if endocytosis of the NRP1 variants, Met and β1-integrin are dependent on dynamin, HCT116 cells were treated with the dynamin inhibitor, dynasore[37], or transfected with a dynamin dominant-negative construct (dynamin-2 K44A-GFP). Treatment with dynasore or expression of the dynamin-2 K44A mutant increased the levels of the NRP1 variants, Met and β1-integrin at the plasma membrane with a concomitant decrease in their intracellular pools (Fig. 7a, b and Supplementary Fig. 9a). Moreover, cell migration and invasion were dramatically inhibited by dynasore in HCT116 cells expressing either of the two NRP1 variants but not NRP1-WT (Fig. 7c, d). Similar results were observed in HCT116 cells knocked down for clathrin heavy chain (CHC) (Fig. 7e–h; Supplementary Fig. 9b). Thus, these results reveal that internalization of the NRP1 variants, Met and β1-integrin and their complex formation to enhance cell migration and invasion are mediated by dynamin- and clathrin-dependent endocytosis.

**The NRP1 variants activate FAK signaling in endosomes.** To determine which pathways are involved in the endosomal NRP1 splice variant/Met/β1-integrin complex-promoted CRC cell dissemination, activation of Met and its downstream PI3K/ AKT, RAS/ERK and FAK/p130Cas signaling cascades, were examined in HCT116 cells expressing NRP1-WT, NRP1-ΔE4,

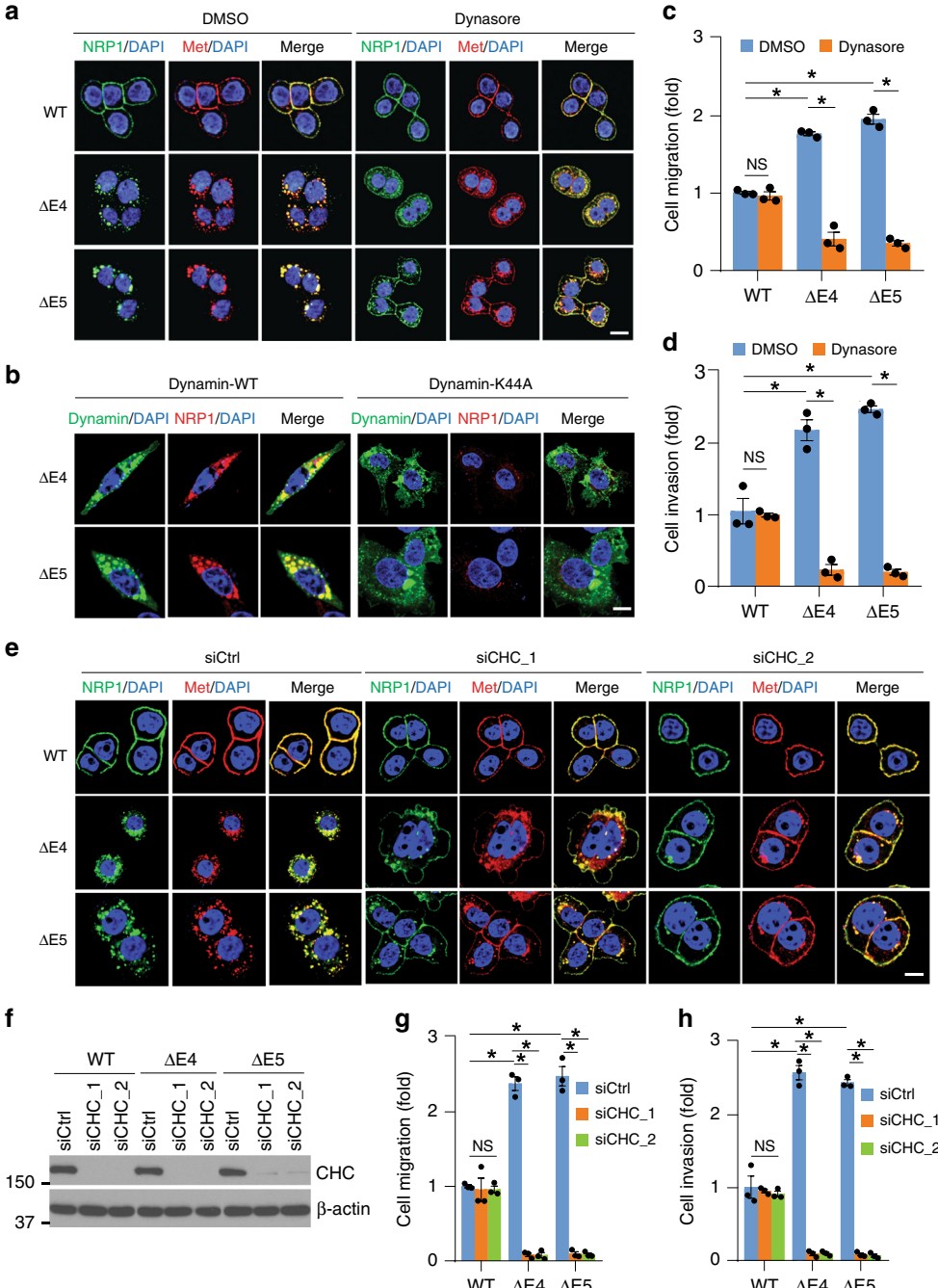

**Fig. 7** Blocking endocytosis of the NRP1 variants inhibits CRC cell migration and invasion. **a** Confocal images of NRP1, Met and DAPI staining in HT29 cells with expression of the indicated NRP1 isoforms that were treated with dynasore (80 μM) or DMSO as control for 2 h. Scale bars, 10 μm. **b** Confocal images of NRP1 with DAPI staining in NRP1-ΔE4- or NRP1-ΔE5-expressing HCT116 cells that were transfected with WT dynamin-2-GFP or dynamin-2 K44A-GFP. Scale bars, 10 μm. **c**, **d** Transwell migration **c** and invasion **d** analyses of HCT116 cells with expression of the indicated NRP1 isoforms in the presence of DMSO or dynasore (80 μM) over 6 h and 30 h of incubation, respectively. The results are expressed as the fold change of cell migration or invasion in the indicated cells relative to the NRP1-WT-expressing cells treated with DMSO. **e** Confocal images of NRP1, Met and DAPI staining in HT29 cells with expression of the indicated NRP1 isoforms that were transfected with two different sets of CHC siRNA or control siRNA. Scale bars, 10 μm. **f–h** HCT116 cells with stable expression of the indicated NRP1 isoforms were transfected with two different sets of CHC siRNA or control siRNA for 36 h, followed by western blot analysis **f**, or by transwell migration **g** or invasion **h** analyses. The results are expressed as the fold change of cell migration or invasion in the indicated cells relative to the NRP1-WT-expressing cells transfected with control siRNA. All graphic data are presented as mean ± s.e.m. ($n = 3$ independent experiments). *$p < 0.001$; NS not significant using Student's $t$-test

NRP1-ΔE5 or vector control. Expression of either NRP1-ΔE4 or NRP1-ΔE5 markedly increased Met phosphorylation levels of Y1234/1235 (kinase domain) and Y1349 (docking site) over that seen for NRP1-WT (Fig. 8a), which suggests enhanced kinase activity of Met. Similarly, the phosphorylation levels of FAK at Y397 and its interacting protein p130Cas at Y249 were also much higher in both NRP1-ΔE4 and NRP1-ΔE5 relative to the NRP1-WT (Fig. 8a). These effects were not due to an increase in the

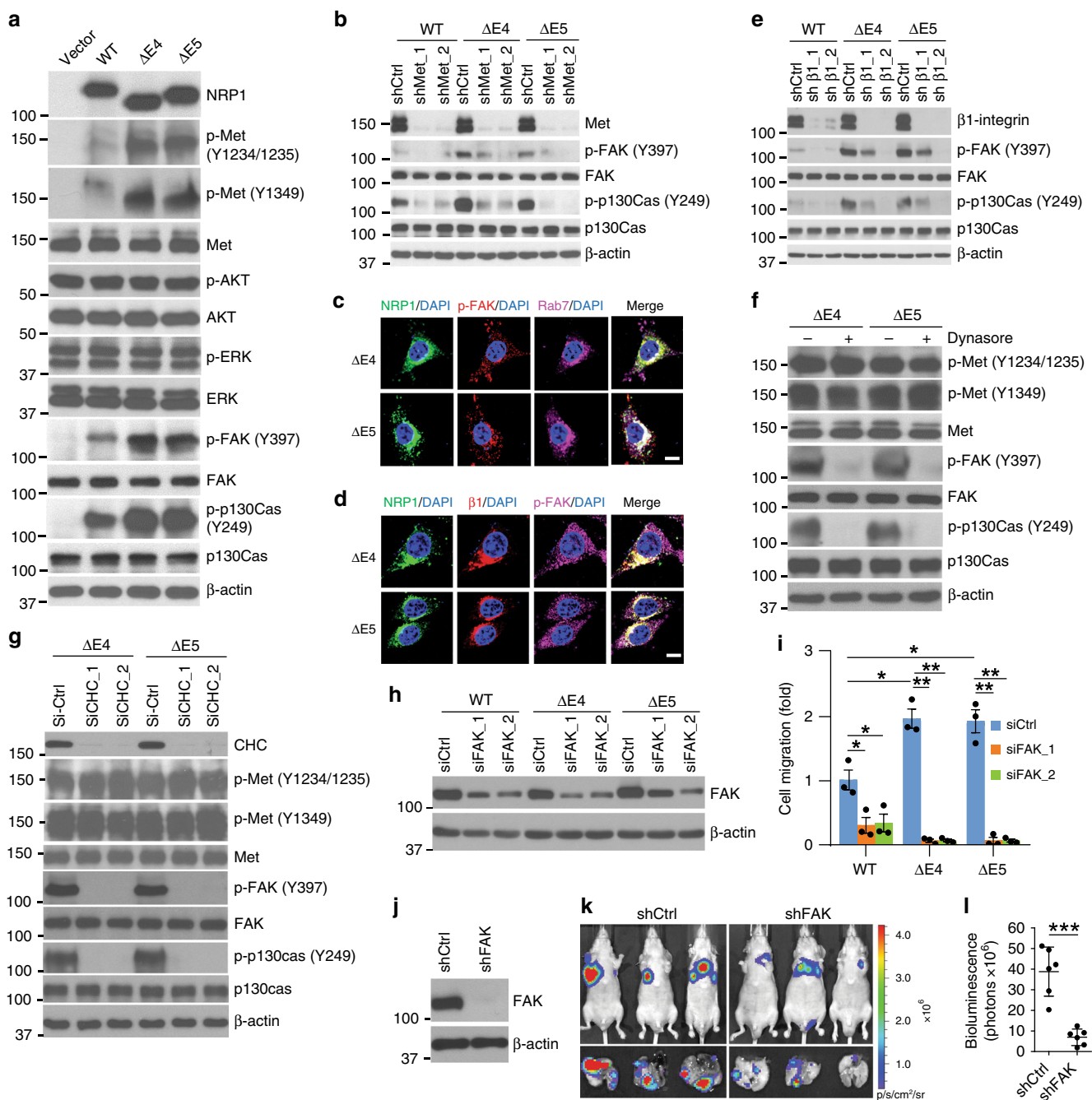

**Fig. 8** The NRP1 variants activate the endosomal FAK signaling for CRC cell dissemination. **a** HCT116 cells with stable expression of vector control, NRP1-WT, NRP1-ΔE4 or NRP1-ΔE5 were assessed by western blot analysis. **b** NRP1-WT-, NRP1-ΔE4- or NRP1-ΔE5-expressing HCT116 cells with stable expression of two different sets of Met shRNA or control shRNA were assessed by western blot analysis. **c, d** Confocal images of NRP1, p-FAK, β1-integrin, Rab7 and DAPI staining in HCT116 cells with expression of the indicated NRP1 isoforms. Scale bars, 10 μm. **e** NRP1-WT-, NRP1-ΔE4- or NRP1-ΔE5-expressing HCT116 cells with stable expression of two different sets of β1-integrin shRNA or control shRNA were assessed by western blot analysis. **f** NRP1-ΔE4- or NRP1-ΔE5-expressing HCT116 cells were treated with DMSO or dynasore (80 μM) for 2 h, followed by western blot analysis. **g** NRP1-ΔE4- or NRP1-ΔE5-expressing HCT116 cells were transfected with clathrin heavy chain (CHC) siRNA or control siRNA for 48 h, followed by western blot analysis. **h, i** HCT116 cells with expression of the indicated NRP1 isoforms were transfected with two different sets of FAK siRNA or control siRNA for 48 h, followed by western blot analysis **h**, or by transwell migration analysis **i**. The results are expressed as the fold change over the migrated cell number found in NRP1-WT cells transfected with siCtrl. Data are presented as mean ± s.e.m. ($n = 3$ independent experiments). *$p < 0.03$; **$p < 0.001$ using Student's $t$-test. **j** Western blot analysis of NRP1-ΔE4-expressing HCT116-Luciferase/GFP cells with stable expression of FAK shRNA or control shRNA. **k** Bioluminescence images of lung metastasis in athymic nude mice that were injected intravenously with NRP1-ΔE4-expressing HCT116-Luciferase/GFP cells with stable expression of FAK shRNA or control shRNA at week 6 post-injection. This experiment was performed by sharing the same shCtrl group with the experimet performed in Fig. 6l, m. **l** Quantitative analysis of bioluminescence in lung metastasis as shown in **k** was performed, and the results are presented as mean ± s.e.m. ($n = 6$ mice/group). ***$p < 0.005$ using Mann–Whitney test

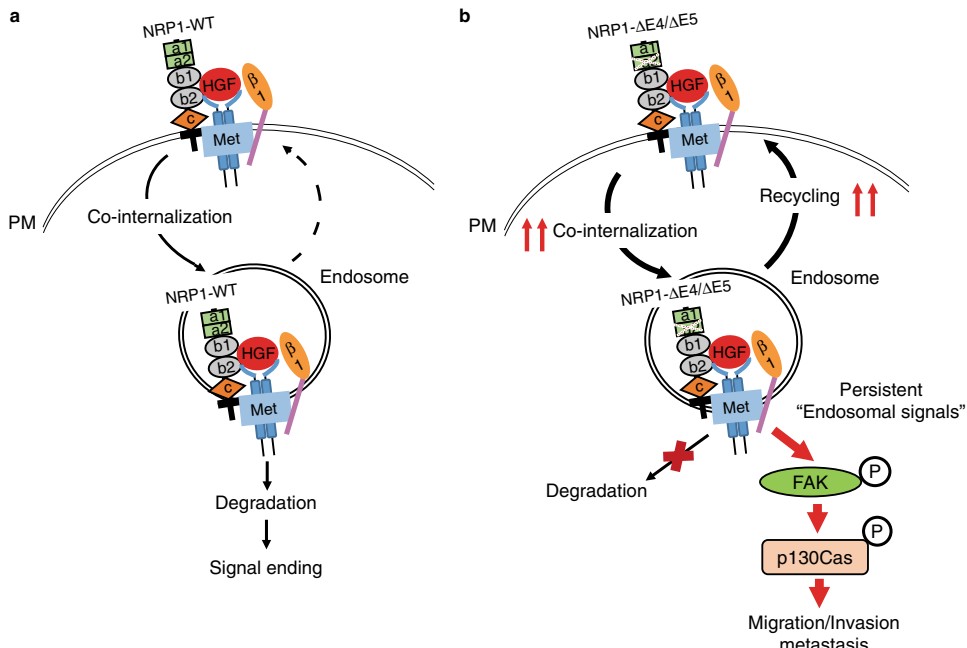

**Fig. 9** A model for NRP1-WT and its splice variants signaling in CRC cell dissemination. **a** NRP1-WT associates with Met and β1-integrin receptors and resides on plasma membrane (PM) under basal conditions, and internalizes upon stimulation by HGF followed by rapid degradation, leading to a restricted period of signal generation. **b** NRP1-ΔE4 or NRP1-ΔE5 renders Met and β1-integrin co-internalization and constitutively recycles back to the cell surface or the extracellular space under basal conditions. These events result in an increased accumulation of the NRP1 variants/Met/β1-integrin complexes on endosomes, which provides persistent endosomal signals to activate the FAK/p130Cas pathway for promoting cell migration, invasion and metastasis in CRC. These oncogenic activities of NRP1-ΔE4 or NRP1-ΔE5 are further enhanced by HGF stimulation

total levels of Met, FAK and p130Cas in the cells expressing NRP1-ΔE4 or NRP1-ΔE5. By contrast, there were no detectable differences in the expression of total protein and phosphorylation levels of AKT and ERK among those cell lines (Fig. 8a). Knockdown of Met expression effectively inhibited phosphorylation of FAK and p130Cas in HCT116 cells expressing either NRP1-ΔE4, NRP1-ΔE5 or NRP1-WT (Fig. 8b). However, complete inhibition of Met phosphorylation by the Met specific inhibitor, PHA-665752[38], did not affect phosphorylation level of FAK, p130Cas, AKT, or ERK (Supplementary Fig. 10a). Consistent with the view that Met internalization depends on its tyrosine kinase activity[39], PHA-665752 did inhibit HGF-induced MET internalization in HCT116 cells (Supplementary Fig. 10b). Nevertheless, PHA-665752 did not affect the co-internalization of the NRP1 variants/Met/β1-integrin complex under basal conditions (Supplementary Fig. 10c). Unlike the marked inhibition of cell migration and invasion by Met knockdown in NRP1-ΔE4- or NRP1-ΔE5-expressing cells (Fig. 6j, k), treatment with PHA-66575 had no such effects (Supplementary Fig. 10d, e). Similar to the reported activation of FAK by endosomal β1-integrin[34], active β1-integrin, phosphorylated FAK and NRP1-ΔE4 or NRP1-ΔE5 co-localized on endosomes (Fig. 8c, d), and silencing β1-integrin also dramatically inhibited phosphorylation of FAK and p130Cas in these cells (Fig. 8e). Furthermore, blocking endocytosis by treatment with dynasore or knockdown of CHC expression did not inhibit phosphorylation of Met, but did decrease the phosphorylation levels of FAK and p130Cas in cells expressing NRP1-ΔE4 or NRP1-ΔE5 (Fig. 8f, g). Collectively, these results indicate that both Met and β1-integrin proteins, but not Met tyrosine kinase activity, are required for endosomal activation of FAK/p130Cas signaling induced by the NRP1 variants.

Silencing FAK using siRNA or treatment with the FAK specific inhibitor VS-6063 that inhibits phosphorylation of both FAK and p130Cas, showed a greater inhibition of cell migration and invasion in HCT116 cells expressing NRP1-ΔE4 or NRP1-ΔE5 than in those expressing NRP1-WT (Fig. 8h, i; Supplementary Fig. 10f-h). Similar results were observed in HCT116 cells with knockdown of p130Cas expression (Supplementary Fig. 10i, j). Additionally, knockdown of FAK expression showed marked suppression of lung metastasis induced by NRP1-ΔE4 in mice (Fig. 8j–l). Thus, the persistent activation of FAK/p130Cas signaling and promotion of CRC cell dissemination are linked directly to enhanced endocytic trafficking of the NRP1 variants/Met/β1-integrin complex and its accumulation and signaling on endosomes (Fig. 9, model).

## Discussion

NRP1 is overexpressed in a range of human cancers including CRC, and increased expression of NRP1 is associated with poor patient prognosis[22,40]. Characterization of the NRP1 variants generated by alternative splicing mechanisms and modification of NRP1 by O-linked glycosylation have provided profound insights into our understanding of the regulation of NRP1 function in cancer growth, angiogenesis, invasion, metastasis and modulation of therapeutic outcomes[5,22,41]. Here, our findings add two human NRP1 splice variants generated by the skipping of exon 4 and exon 5, respectively, in CRC. While the NRP1-ΔE4 is dominantly expressed in CRC associated with tumor progression, both NRP1 splice variants are defective in N-linked glycosylation modification, and exhibit increased endocytosis/recycling activity with decreased levels of degradation, which lead to their accumulation on endosomes and persistent activation of FAK/p130Cas signaling through interaction and co-internalization with Met and β1-integrin. This perturbed trafficking of the two NRP1 splice variants promotes CRC cell migration, invasion and metastasis, directly linking endocytosis and metastatic progression (Fig. 9).

Ligand-induced endocytosis of signaling receptors is an important mechanism for negatively regulating signaling from the cell surface through rapid internalization and subsequent transport to late endosomes and lysosomes for degradation[15]. However, a growing body of evidence indicates that receptors such as EGFR and Met receptor tyrosine kinases (RTKs) or β1-integrin can remain active within endosomes to provide spatially and temporally restricting signals that contribute to pathway-specific tumor progression[16,17,34,42–44]. NRP1 lacks a typical kinase domain, primarily functioning as a co-receptor to form ligand-specific receptor complexes[7]. Our data are consistent with the view[10] that NRP1-WT associates with the Met receptor and resides on the plasma membrane, and that they co-internalize upon stimulation by HGF, but not VEGF, followed by rapid degradation in CRC cells (Fig. 9a). It is noteworthy that NRP1-ΔE4, NRP1-ΔE5 and the N-glycosylation-defective NRP1 mutants N150Q and N261Q also localized on cell surface 24 h after transient or stable expression under serum-free conditions. These data suggest that altered N-glycosylation modification does not affect NRP1 transport to the plasma membrane after its synthesis. However, HGF stimulation even with low levels of HGF under basal conditions can markedly induce endocytosis of the NRP1 splice variants or N150Q and N261Q mutants together with Met and β1-integrin receptors (Fig. 9b). Unlike NRP1-WT, the NRP1 splice variants display enhanced binding capabilities with both Met and β1-integrin to form the NRP1 variants/Met/β1-integrin complexes, leading to co-internalization and co-accumulation on late endosomes. These endosomal complexes provide persistent signals to activate the FAK/p130Cas pathway, and thereby, promote CRC cell migration, invasion and metastasis. These effects were not induced by $VEGF_{165}$ stimulation. Structurally, the b1/b2 domain of NRP1 is required for binding to $VEGF_{165}$, whereas the presence of a1/a2 domain in NRP1 enhances $VEGF_{165}$ binding considerably[45]. NRP1 can also bind to HGF, which is likely mediated through its b1/b2 domain[28]. Given our findings that (1) NRP1-ΔE4 and NRP1-ΔE5 lack a part of a2 domain with defective N-glycosylation modification at the sites N150 and N261, respectively, and had modest internalization upon $VEGF_{165}$ stimulation; and (2) the other three putative N-glycosylation-defective mutants, N300Q, N522Q and N842Q, or the O-linked glycosylation-deficient S612A mutant did not affect NRP1 localization on the plasma membrane and were less responsive to HGF stimulation, we propose that the partial deletion of a2 domain or defect in N150- or N261-linked glycosylation modification may reduce NRP1 binding to $VEGF_{165}$, but enhance its binding to HGF, which results in the observed enhancement in HGF-stimulated endocytosis even in cells with depletion of Met or β1-integrin under basal conditions (Fig. 6f, g). Although this notion requires further investigation, the current study reveals a critical role for N150- or N261-linked glycosylation modification in the regulation of HGF-stimulated NRP1 endocytosis and activation of endosomal signaling that contribute to CRC metastatic spread. The NRP1 S612A mutant, deficient in O-linked glycosylation, was previously shown to enhance cell invasion of U87MG glioma cells by increasing p130Cas phosphorylation, but the underlying mechanism is unclear[26]. Interestingly, we found that similar to NRP1-WT, the NRP1 S612A mutant was unstable compared to the NRP1 N150Q or N261Q mutants (Supplementary Fig. 4c, d). Whether a defect in N-linked glycosylation of NRP1 is more aggressive than in O-linked glycosylation of NRP1 to promote metastatic progression is another matter that remains to be further characterized.

The enhanced level of endocytosis/recycling of the two NRP1 metastasis-promoting variants, along with their impaired degradation, indicates that they undergo a persistent shuttling between the cell surface and endosomes. Such shuttling of ligand-stimulated RTKs and integrins has been shown to regulate the spatial restriction of signaling necessary for directed migration[17,46,47]. Although a small amount of the cell surface-biotinylated NRP1-ΔE4 and NRP1-ΔE5 was detected using the streptavidin-agarose pulldown analysis (Fig. 2d), these variants were barely detected on plasma membrane under basal conditions by immunofluorescence analysis (Fig. 2b, c). These data suggest that the two NRP1 variants are rapidly internalized after recycling back to the cell surface and/or they may transport to the extra-cellular space by secreted exosomes. Exosomes are vesicles derived from late endosomes, also known as multivesicular bodies, and can be secreted to the extracellular environments by most cell types[48,49]. Indeed, we found that the two NRP1 variants also co-localized with several common exosome marker proteins, including CD63, CD81 and Alix[50] (Supplementary Fig. 11). Exosomes are key mediators of intercellular communication and can be detected in the tumor microenvironment. Emerging evidence suggests that exosomes play an important role in facilitating tumorigenesis by regulating angiogenesis and metastasis[48,51]. Thus, it is likely that the NRP1 variants are secreted by exosomes and continuously internalize and sustain the active NRP1 variants/Met/β1-integrin complexes on endosomes, and thereby, provide persistent endosomal signals for tumor progression.

The increased endocytotic activity of the NRP1 variants/Met/β1-integrin complexes is dependent on GIPC1, clathrin and dynamin, but is independent of Met activity, because the internalization of these complexes was not affected by a Met-specific tyrosine kinase inhibitor. Silencing Met or β1-integrin also does not affect internalization of the NRP1 splice variants, but depletion of the NRP1 variants blocks endocytosis of both Met and β1-integrin. While regulation of the endocytosis of the NRP1 variants/Met/β1-integrin complexes need further characterization, our results strongly suggest that the physical interaction with the NRP1 variants and NRP1 adaptor protein GIPC1 are required for co-internalization of Met and β1-integrin under the basal condition with low levels of HGF. On the other hand, the NRP1 variants also require both Met and β1-integrin as partners in endosomes for sustained activation of the FAK/p130Cas signaling pathway to promote CRC cell dissemination. While Met has been shown to directly bind FAK leading to FAK activation in a Met phosphorylation-dependent manner[52], active β1-integrin can also positively regulate FAK activity on endosomes[34]. In addition, several studies highlight an important role for the cross-activating Met/β1-integrin complex in promotion of cancer invasion and metastasis[53–55]. Our current study shows that pharmacological inhibition of Met had no effect on FAK activation and cell migration and invasion in CRC cells expressing the NRP1 variants and suggests that β1-integrin bypass signaling contributes to FAK activation and renders Met tyrosine kinase inhibitor-resistant in these cells. Indeed, silencing β1-integrin profoundly inhibited FAK/p130Cas signaling for CRC cell motility and metastasis, and similar results were observed for Met knockdown that caused active β1-integrin to dissociate from endosomal NRP1 variants. Thus, our findings highlight the functional importance of the NRP1 variants/Met/β1-integrin complexes on endosomes in activating FAK signaling for CRC metastasis. Strikingly, blocking endocytosis of these complexes by pharmacological or genetic blockers of endocytosis or by silencing clathrin expression in the NRP1 variant-expressing cells markedly decreased FAK/p130Cas signaling as well as the migratory and invasive phenotypes. Similar results were observed by genetic or pharmacological inhibition of FAK/p130Cas signaling. Thus, strategies aimed at blocking endocytosis or formation of NRP1 variants/Met/β1-integrin complexes or alternatively inhibiting their endosomal signals on activation of FAK/p130Cas pathway

using pharmacological inhibitors of FAK, may have therapeutic potential in CRC and possibly other types of cancers with expression of the N-glycosylation-defective NRP1 splice variants.

## Methods

**Cell lines and human CRC specimens.** Human HCT116, HT29 and DLD-1 cell lines were obtained from American Type Culture Collection (ATCC). HCT116 and HT29 cells were cultured in McCoy's medium (Sigma). DLD-1 cells were cultured in RPMI-1640 medium (Sigma). Primary human CRC cell lines, Pt93 and Pt130, as well as Pt2337 and its paired liver metastasis cell line, LM2377, were established from patient primary and metastatic tumors (Markey Cancer Center, University of Kentucky)[56]. The primary CRC cells were cultured in DMEM medium (Sigma). All media were supplemented with 10% FBS (Sigma), streptomycin (100 μg ml$^{-1}$) and penicillin (100 units ml$^{-1}$). All cell lines were subjected to regular mycoplasma testing via PCR using e-Myco Plus kit (iNtRON Biotechnology) and underwent short tandem repeat (STR) profiling (Genetica). The 126 frozen fresh primary colorectal tumors including NF90, NF99, NF103, NF105, NF106, and NF110 and adjacent normal control tissues (N1-4) were collected from patients who had undergone surgery resections at the Nanfang Hospital, Southern Medical University, China and the Markey Cancer Center, University of Kentucky. Experiments were performed under protocols approved by the Southern Medical University Ethics Committee and the University of Kentucky Institutional Review Board. All CRC cases were confirmed by a senior pathologist and staged based on the 2011 Union for International Cancer Control TNM classification of malignant tumors. The pathological diagnoses of all enrolled patients were confirmed by two different pathologists according to the WHO grading system[57]. Data regarding the clinical characteristics of patients are listed in Supplementary Tables 1 and 2.

**Growth factors and chemicals.** VEGF$_{165}$, EGF, and HGF were purchased from R&D Systems. LysoTracker Deep Red was purchased from ThemoFisher Scientific. Cycloheximide, tunicamycin, swainsonine, benzyl 2-acetamido-2-deoxy-α-D-galactopyranoside (BADGP) and dynasore were obtained from Sigma. PHA-665752 was purchased from Selleckchem. VS-6063 was purchased from MedChem Express.

**RT-PCR.** Total RNA was isolated from cells using the RNeasy plus mini kit (Qiagen), and from tissues using TRIzol reagent (ThermoFisher Scientific), according to the manufacturer's instructions. Equal amounts of RNA were used as templates for all reactions. cDNA was generated with the SuperScript III First Strand Synthesis System (ThermoFisher Scientific). PCR was performed using the Phusion Hot Start II High-Fidelity DNA Polymerase kit (ThermoFisher Scientific). The primers 5′-TACGAAACACATGGTGCAGGA-3′ (forward) and 5′-CTGCAG ACCAGTTGGTGCTAT-3′ (reverse) were used to amplify NRP1-WT, NRP1-ΔE4 and NRP1-ΔE5 in the human samples. The amplified PCR fragments were separated on an agarose gel and subjected to DNA sequencing. Primers 5′-ACAACTT TGGTATCGTGGAAGG -3′ (forward) and 5′-GCCATCACGCCACAGTTTC-3′ (reverse) were used to amplify human GAPDH.

**Plasmids.** Human NRP1-WT, NRP1-ΔE4 and NRP1-ΔE5 were amplified by PCR using a HCT116 cDNA library, and then subcloned into the pLenti6.3 or pCMV6 vector[58,59]. The NRP1 mutants N150Q, N261Q, N150Q/N261Q, N300Q, N522Q, N842Q and S612A were generated using a QuikChange XLII mutagenesis kit (Stratagene). All sequences were verified by automated DNA sequencing. To establish stable transfectants with specific protein expression, cells were infected with lentivirus using the indicated pLenti6.3 constructs followed by selection with puromycin (2 μg ml$^{-1}$) for 7–10 days[58]. Dynamin-2 K44A-GFP (#34687) and WT dynamin-2-GFP (#34686) were purchased from Addgene.

**Gene silencing by siRNA and shRNA.** The siRNA against human NRP1 (#79833, #79835), clathrin (#34735, #34736), FAK (#34733, #34734), p130Cas (#34731, #34732) and non-targeting control siRNA (#0207) were obtained from Gene-Pharma. The ON-TARGETplus human GIPC1 siRNA pool (L-019997-00-0005) and non-targeting control siRNA pool (D-001810-10) were purchased from Dharmacon. Cells were transfected with 50 nM siRNA against the indicated genes or control siRNA using Lipofectamine RNAiMAX reagent according to the manufacturer's instructions (ThermoFisher Scientific). After 36–48 h transfection, cells were utilized for the indicated assays. The lentiviral shRNAs against human NRP1, Met, β1-integrin or FAK were cloned into the pLKO.1 vector (Sigma). The Non-Target Control shRNA (SHC002) was from Sigma. Both siRNA and shRNA sequences are listed in Supplementary Table 3. To establish stable transfectants with knockdown of specific protein expression, cells were infected with lentivirus using the indicated shRNA constructs followed by selection with hygromycin (250 μg ml$^{-1}$) for 7–10 days[58].

**Immunofluorescence and confocal microscopy analysis.** Cells ($5 \times 10^4$) were grown on collagen-precoated coverslips in the regular growth medium containing 10% FBS, or serum starved overnight followed by stimulation with VEFG$_{165}$

(50 ng ml$^{-1}$), EGF (50 ng ml$^{-1}$) or HGF (50 ng ml$^{-1}$) for 30 min. Subsequently, cells were fixed with 4% paraformaldehyde in PBS for 15 min, permeabilized in 0.2% Triton X-100 and 0.5% BSA in PBS for 5 min and then blocked with 4% BSA in PBS for 10 min. The cells were incubated overnight at 4 °C with the indicated primary antibody. After three washes with 0.05% Triton X-100 in PBS, cells were incubated with the indicated secondary antibody for 1 h. Cells were then washed three times, mounted with DAPI containing mounting medium (H-1200, Vector Laboratories), viewed, and photographed under a Nikon A1⁺-Ti2 confocal microscope. For image quantifications, picture fields were chosen arbitrarily on the basis of DAPI staining. For the NRP1 variant and mutant intracellular localization and co-localization experiments, ten pictures were taken per condition. The percentages of co-localizations were determined using Nikon NIS-Elements AR software. The primary and secondary antibodies used in this assay are listed in Supplementary Table 4.

**Immunoprecipitation and western blot analysis.** Cells were lysed in RIPA buffer (50 mM Tris-HCl, pH 7.5, 150 mM NaCl, 1 mM EDTA, 1% NP-40, 0.1% SDS, 0.5% sodium deoxycholate, 10% glycerol, protease and phosphatase inhibitor cocktails). Protein concentrations were measured using the BCA protein assay reagent (Thermo Fisher Scientific). The cell lysates (500 μg protein) were immunoprecipitated with 2 μg of the indicated antibody overnight followed by incubation with a 50% slurry of protein G sepharose beads for 3 h at 4 °C. The beads were washed three times with the lysis buffer, and the immunoprecipitated protein complexes were resuspended in 2× Laemmli sample buffer followed by western blot analysis. For western blot analysis, equal amounts of protein were resolved by SDS-PAGE, transferred to PVDF membranes, immunoblotted with specific primary and secondary antibodies, and detected using chemiluminescence (GE Healthcare). A list of all antibodies used in this work and dilutions can be found in Supplementary Table 4. Contrast of western blot images was adjusted using Adobe Photoshop and uncropped and unprocessed scans are found in the Source Data file.

**Detection of NRP1 N-glycosylation.** Cells were lysed in RIPA buffer, and the cell lysates were treated with PNGase F (peptide N-glycosidase F) for 3 h at 37 °C according to the manufacturer's instructions (#P0704, New England Biolabs). The reaction was stopped by the addition of 5 × loading buffer followed by western blot analysis using anti-NRP1 antibody (#3725, Cell Signaling Technology).

**Cell surface biotinylation assay.** To measure the relative proportions of the cell surface and internal pools of NRP1, cell surface proteins were labeled covalently using a membrane-impermeable biotinylation reagent (N-hydroxysulphosuccinimide (sulpho-NHS)-SS-biotin; ThermoFisher Scientific) according to the protocol described by Joffre et al.[17]. Briefly, cells were incubated with 0.15 mg ml$^{-1}$ biotin for 10 min at 4 °C and the excess biotinylation reagent was quenched by washing with the cold buffer containing 25 mM Tris at pH 8, 137 mM NaCl, 5 mM KCl, 2.3 mM CaCl$_2$, 0.5 mM MgCl$_2$ and 1 mM Na$_2$HPO$_4$. Cells were lysed in RIPA buffer and centrifuged. A fraction of the supernatant (total cellular NRP1 or Met = 'total') was collected. The residual supernatant was added to prewashed streptavidin-agarose beads (ThermoFisher Scientific) and rotated at 4 °C for 2 h. Beads were collected by centrifugation and the supernatant (internal pool of NRP1 or Met = 'unbound') was collected. Beads were washed 3 times with the RIPA buffer at 4 °C and proteins (surface pool of NRP1 or Met = 'bound') were extracted by heating at 95 °C with sample buffer. Equivalent volumes were analyzed by western blot and densitometric analyses were carried out. The percentage of intracellular NRP1 or Met was calculated using the formula: intracellular NRP1 or Met receptor = (NRP1 or Met in unbound fraction)/(total NRP1 or Met) × 100.

**Biotinylation internalization and recycling assay.** The trafficking assay for NRP1 and its variants was performed according to the method described by Joffre et al.[17]. Cells were cultured in serum-free medium overnight followed by washing with cold PBS. On ice, cell surface proteins were labeled with 0.2 mg ml$^{-1}$ NHS-SS-biotin in PBS for 45 min. Labeled cells were washed with cold PBS and incubated at 37 °C in culture medium containing 10% FBS, to allow protein trafficking. At the indicated times, the medium was aspirated and the dishes were transferred to ice and washed with cold PBS. Biotin was removed from proteins remaining at the cell surface by reduction with 180 mM of the membrane-impermeable reducing agent MesNa (sodium 2 mercaptoethane sulphonate, Sigma) in 50 mM Tris and 100 mM NaCl at pH 8.6 for 15 min. MesNa was quenched by the addition of 180 mM iodoacetamide (IAA, Sigma) for 10 min.

To measure internalized NRP1 and its variants, the assay was terminated by lysing the cells. Lysates were passed three times through a 27-gauge needle and clarified by centrifugation (17,000 g). Equal protein amounts were incubated with streptavidin-agarose beads with agitation at 4 °C for 2 h. The beads were collected by centrifugation (7,000 g), washed in lysis buffer and proteins were extracted by heating at 95 °C with sample buffer.

To measure the proportion of internalized NRP1 proteins and the variants that recycle back to the cell surface, the internalized fraction was returned to 37 °C for 15 min as a chase. Cells were then returned to ice and biotin was removed from recycled proteins by a second reduction with MesNa. Unreacted MesNa was

quenched with IAA and cells were lysed and processed as before for measurements of internalized receptor.

For each internalization or recycling assessment, two controls were carried out. To measure the total NRP1 or its variants at the surface, biotinylated cells at 4 °C were lysed without biotin reduction. To verify the efficiency of the surface biotin removal, the biotin reduction and MesNa quenching steps were carried out on cells that had remained on ice (time 0) and lysis was carried out.

Equivalent volumes were analyzed by western blotting of NRP1 and densitometric analyses. The percentages of internalized or recycled NRP1 and its variants were calculated using the following formulae: internalized receptor = (NRP1 or its variant level after first incubation at 37 °C)− (NRP1 or its variant level at time 0)/(total surface NRP1 or its variant) × 100; recycled receptor = 100− [(NRP1 or its variant level after 15 min of re-incubation at 37 °C)− (NRP1 or its variant level at time 0)/(NRP1 or its variant level after first incubation at 37 °C)− (NRP1 or its variant level at time 0) × 100].

**Enzyme-linked immunosorbent assay (ELISA)**. The concentration of HGF in FBS and the FBS neutralized with bovine HGF antibody (NBP2-12355, Novus Biologicals) was determined in triplicate wells using the Bovine HGF ELISA Kit according to the manufacturer's protocol (LifeSpan BioSciences).

**Cycloheximide chase assay**. Cells were treated with cycloheximide (50 μgml$^{-1}$) and harvested at indicated time points. The cells were lysed in RIPA buffer and equal amounts of total protein were analyzed by western blotting.

**Cell growth assay**. Cells ($5 \times 10^4$/well) were seeded in 6-well plates in triplicate. The number of viable cells was counted every day for 3 days using the Vi-CELL XR 2.03 (Beckman Coulter)[58,60]. Each experiment was performed in triplicate and repeated at least three times.

**Rac1 activity assay**. Rac1 activity was assessed using the glutathione S-transferase (GST)-tagged p21 binding domain of PAK1 (GST–PBD) pulldown assay[61]. Briefly, cells ($2.5 \times 10^6$) were grown on collagen-precoated 6 cm dishes to ~70% confluency in regular growth medium containing 10% FBS, or serum starved overnight followed by stimulation with HGF (50 ngml$^{-1}$) for 5 min. Subsequently, the cells were lysed in lysis buffer (50 mM Tris, pH 7.4, 100 mM NaCl, 1% NP-40, 10% glycerol, 2 mM MgCl$_2$, and protease and phosphatase inhibitor cocktails), and soluble proteins were incubated with purified GST-PBD to pull down activated Rac1. The amount of total and active Rac1 was detected using Rac1 antibody (Supplementary Table 4).

**Cell migration and invasion assays**. Migration and invasion assays were performed in Boyden chambers coated with collagen or Matrigel, respectively[60], as instructed by the manufacturer (BD Biosciences). Briefly, cells were added to the upper chamber of the transwell insert. Complete medium containing 10% FBS, HGF (50 ngml$^{-1}$) or VEGF$_{165}$ (50 ngml$^{-1}$) as a chemoattractant was added to the bottom chamber. To assess the effect of endocytosis and kinase inhibitors on cell migration and invasion, the inhibitors or their vehicle control DMSO were added to the bottom chamber medium. The plates were incubated at 37 °C in 5% CO$_2$ for indicated time periods. After incubation, cells in the upper compartment were removed with a cotton swab, and cells that migrated or invaded to the filter surface facing the bottom chamber were fixed in 4% paraformaldehyde and stained with 0.2% crystal violet. The number of migrated or invaded cells were counted in at least five areas at x 20 magnification using an inverted microscope.

In addition to the transwell migration assay, a time-lapse live cell imaging system was used to record the migration of live cells. Cells were placed on collagen-precoated glass-bottom culture dishes and monitored at 37 °C using a Nikon BioStation IMQ equipped with a CO$_2$ incubation chamber. Time-lapse phase images were taken every 30 min for 24 h. The movement of twenty individual cells was tracked and analyzed using Nikon Element AR software.

**Animal studies**. Male athymic nude mice (6 weeks old) were purchased from Taconic (Hudson, NY) and maintained and treated under specific pathogen-free conditions. Experiments were carried out under a protocol approved by the University of Kentucky Institutional Animal Care and Use Committee. For the experimental lung metastasis assay, cells with co-expression of firefly luciferase and GFP were injected into the tail vein ($1 \times 10^6$/mouse) of athymic nude mice ($n = 6$/group) as described[60]. To monitor metastasis, mice and lung tissues were imaged with luciferase signals using the IVIS Spectrum system and results were analyzed by the Living Image 3.0 software (Caliper Life Science).

**Statistical analysis**. Statistical analyses for each experiment were performed as described in the corresponding figure legends. Data between groups were compared using a two-tailed unpaired Student's t-test, Mann–Whitney test or χ$^2$-test. All data are presented as mean ± s.e.m. Differences between groups were considered statistically significant at $p < 0.05$. GraphPad Prism software was used for these analyses.

**Reporting summary**. Further information on research design is available in the Nature Research Reporting Summary linked to this article.

## Data availability

All data supporting the findings of this study are available within the article and its supplementary information files and from the corresponding author upon reasonable request. A reporting summary for this article is available as a Supplementary Information file. The source data underlying Figs. 1d–f, 2a, d, e, g–i, 3c–h, 4a, b, f, 5b, c, e, f, h, 6a, b, d, e, h, j, k, m, 7c, d, f–h and 8a, b, e–j, l and Supplementary Figs. 2a, b, e, h, 3c, e, g, i, j, 4c, d, 5a–d, f, g, 6a–c, g–i, k–m, 7c–g, 8a–d and 10a, d–j are provided as a Source Data file.

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

## Acknowledgements

We thank the Markey Cancer Center's Research Communications Office for assistance with manuscript preparation, Emilia Galperin for helpful discussions, and Qun Yan, Yanying Nong and Yuqian Zhou for helpful collection of colorectal cancer specimens. The authors also acknowledge the assistance of the Biospecimen Procurement and Translational Pathology Shared Resource Facility of the Markey Cancer Center (P30CA177558). This work was supported by NCI grant R01CA203257, start-up funds, and pilot grants from CCSG P30CA177558 (UK Markey Cancer Center) and CCTS UL1TR001998 (University of Kentucky). This work was also supported in part by funding from Guangdong Gastrointestinal Disease Research Center (No. 2017B02029003 to S.L.).

## Author contributions

X.H. performed most of the experiments and analyzed the data. Q.Y. performed the animal study, some experiments and analyzed the data. M.C. performed Rac1 activity assay. A.L., W.M., Y.F. and Y.Y.Z. provided essential reagents. K.L.O. and C.W.V.K. discussed the data, provided comments and edited the manuscript. Q.-B.S. and S.L. conceived and designed the study, and supervised data analysis. Q.-B.S. wrote the manuscript.

## Additional information

**Competing interests:** The authors declare no competing interests.

