## [Peer Review File · Nature Communications]

Reviewers' comments:

Reviewer #1 (Remarks to the Author):

This is an interesting manuscript, suggesting the potential role of two specific glycosylation sites of NRP1 in c-Met driven metastatic progression of colorectal cancer. The experiments are well designed and mostly in agreement with the interpretation of the authors. This reviewer has the following comments.

Comments:

1. What percentage of colorectal cancer tissues has these specific exon 4 and 5 deletions of NRP1. Do exon 4 and 5 deleted NRP1 (especially when consider along with cMET) correlate with colorectal cancer progression.
2. Fig 2: For Fig B and C, immunohistochemistry should be repeated with a known cell membrane protein. Z-stack image should be provided to confirm intracellular localization. It seems that exon 4 of NRP1 has been deleted in both the Pt2377 and LM2377. No result for endogenous deletion of exon 5 in colorectal cancer tissue has been provided, although the authors have mentioned such result in the text of the manuscript.
3. Quantitation of supplement figure S3b and 3e should be provided. This experiment should be repeated with other lysosomal markers. This is an important information, which would provide that these splice variants are not delivered to lysosomes for degradation.
4. Figure 6C is interesting as it is suggesting that the internalization of c-Met is entirely dependent of NRP1. I am not sure whether this experiment has been performed in the presence of HGF. Do the authors mean that without NRP1 there will be no ligand-induced internalization of c-Met.
5. Endosomal C-Met mediated Rac1 signaling has been reported for enhanced cell migration. Sustained Rac1 activation therefore should be the read out for the endosomal cMet signaling and cell migration.

Reviewer #2 (Remarks to the Author):

This study from Huang and co-authors investigates the function of two isoforms of the transmembrane receptor Neuropilin-1 (NRP1) generated by exon skipping in colorectal cancer cells. In these isoforms, N-glycosylation is defective and causes aberrant trafficking of the NRP1 variants that are internalized and constitutively recycled to the cell surface, escaping the degradative pathway.

NRP1 is well known to interact with Receptor Tyrosine Kinases (RTKs), including MET, and regulate endocytosis of receptors on the cell surface. The Authors found here that an increased association of these novel NRP1 mutants with MET promotes their concomitant internalization, resulting in MET accumulation in endosomes. This is associated with persistent activation of the FAK/p130Cas signaling pathway, ultimately promoting invasion and metastatic dissemination of colon cancer cells.

The quality of the data presented here is good and the statistical analysis meaningful, even if a non-parametric tool such as the Mann-Whitney test would have been more appropriate than a two-tailed Student t-test. Investigation of receptor trafficking has been performed using a wealthy of tools and approaches generating convincing results. Methods are adequately described. However, the preliminary evidence that inhibition of MET kinase phosphorylation by PHA-665752 does not prevent FAK activation induced by NRP1 variants, nor the downstream effects attributed to MET function, poses the question about the actual implicated mechanisms, which are not elucidated here.

Major points:

1) According to the proposed working model, NRP1 variants promote MET endocytosis in HGF-stimulated cells. However, NRP1 isoforms and MET are also internalized under basal conditions in the presence of FBS (Figure 2b,c and Figure 6f), but not in starved cells (Figure 2f). Is this also accounted by HGF-dependent MET activation? or by the involvement of other factors? Since HGF-induced MET internalization is known to depend on its tyrosine kinase activity, one would expect that NRP1 variants-associated endocytosis were also dependent on MET kinase activity, while this was surprisingly excluded by data shown in Suppl. Fig. 7b. How can this apparent conflict be explained?

2) The functional impact of NRP1 variants in cancer cell migration and invasion is shown to be dependent on MET and FAK expression, by means of RNAi-based experiments (in Fig. 6i-j and 8f). However, the relevance of this pathway associated with the expression of NRP1 variants should be validated also for the induction of metastatic spreading in vivo, which is particularly relevant in translational perspective.

3) The presented experiments are almost always based on overexpression of the NRP1 variants in cancer cells. It is understandable the difficulty to target specifically the endogenous splice variants without affecting the wild type transcript. However, some data should be provided to validate the relevance of endogenously expressed NRP1 variants in cancer cells. For instance, the Authors could assay their preferential co-immunoprecipitation with MET, as claimed based on data shown in Figure 6a. In fact, Fig. S2c shows that the endogenous variants can be detected and clearly discriminated from WT-NRP1 by western blotting.

4) NRP1 variants-induced MET internalization is proposed to activate FAK signaling, and MET downregulation by RNAi can block the pathway. However, preliminary data shown in Supplementary Figure S7a seem to suggest that FAK activation here is not due to MET kinase activity, as it is not blocked by PHA-665752 inhibitor. Actually, direct interaction between MET and FAK has been reported previously, and indeed it depends on MET phosphorylation (DOI: 10.1128/MCB.02186-05). This is in contrast with findings presented here and challenges the relevance of data shown in Figure S7, as no alternative signaling mechanism is proposed to account for FAK regulation by MET in this study. In general, it is puzzling that the treatment with an inhibitor of MET kinase activity, which is known to be crucial for HGF-induced regulation of cell migration and invasion, is unable to block the downstream effects induced by NRP1 variants and mediated by MET expression. At the present stage, I would recommend to reconsider the relevance to these data for the study. One option could be to leave out Figure S7, acknowledging that the mechanism of FAK activation by MET in cells overexpressing NRP1 variants needs to be further investigated. In any case, the involvement of MET in this pathway independent of its kinase activity should be accurately discussed, in face of the current literature.

5) Notably, NRP1 has been shown also to interact with Integrins (doi: 10.1038/s41388-018-0290-4), which are major regulators of FAK activation in endosomes (doi: 10.1038/s41388-018-0290-4, doi: 10.1038/ncb3250, doi: 10.1016/j.tcb.2016.02.001). Therefore, the possibility that, along with MET, integrins are constitutively internalized by NRP1 variants and activate FAK on endosomes should be addressed.

6) Endosomal activation of FAK has been demonstrated in several studies. The authors suggest that, in the colon cancer cell lines analyzed here, FAK activation is due to persistent MET signaling from endosomes, which, in turn, is driven by NRP1 variants. However, visualization of FAK activation on endosomes positive for NRP1 isoforms would strengthen the proposed model.

Additional points:

7) Semi-quantitative analysis of NRP1 variants expression shown in Fig. 1e should be validated by comparison to the levels of a housekeeper gene mRNA in the same samples. It is surprising that normal colonic mucosa does not express any NRP1 transcript.

8) Concerning results shown in Figure 2, a western blot revealing VEGFR, EGFR and MET protein levels in the cell lines under analysis should be provided, in order to assess the relative expression of these receptors.

9) With reference to data shown in Fig. 3d-e, in midpage 7, it is said "After 15 min incubation with culture medium"; it should be specified what kind of medium was used here? FBS-containing or serum-free?

10) The relevance of data shown in Supplementary Fig. 5f-g is questionable. In fact, how can the impact of WT vs. splice variant NRP1 be discriminated here? Indeed the cells express all mRNA isoforms. This major limitation should at least be highlighted in the manuscript.

11) Data plotting in Fig. 8f is not appropriate, as the Y-axis scale is severely cut to values above 60%, and the basal impact of FAK silencing on migration is not reported. Absolute scores of cell migration should be plotted, or % inhibition of controls by showing the entire range 0-100%.

12) In the study, it is also investigated the potential regulatory role of NRP1 variants on EGFR tyrosine kinase. This is indeed justified by previous literature linking wild type NRP1 to EGFR in other experimental models (e.g. DOI: 10.1158/0008-5472.CAN-12-0995). Even if the results shown here in colon cancer cells do not confirm this putative working hypothesis, in order to put the data into context, the relevant previous literature should be cited.

13) The text needs to be carefully edited for typos. For instance, in page 4 line 8: "Furthermore, our work highlight that..."; or in page 7 line 9: "may secret to the extracellular space..."; etc...

Reviewer #3 (Remarks to the Author):

This manuscript from Qing-Bai She and colleagues reports the discovery of neuropilin-1 (NRP1) splice variants in colorectal cancer. These two NRP1 variants are shown to constitutively internalise, recycle and to be protected from degradation. It is further shown that these mutants miss a distinct glycosylation site each. Mutants of either of these glycosylation sites phenocopy the splice variant mutants for the constitutive internalisation and increased stability. Splice variants and glycosylation mutants enhance cell migration, invasion and in vivo lung colonisation. Co-immunoprecipitation and colocalization in vesicles between Met receptor and the NRP1 splice variants are observed. Moreover, Met internalisation and stability are increased in cells expressing NRP1 splice variants versus cells expressing NRP1 WT. It is further shown that blocking endocytosis with dynasore, clathrin knock down or expression of dynamin dominant negative inhibit NRP1 variants and Met co-internalisation and cell migration and invasion. Met was found phosphorylated in the cells expressing NRP1 variants with a parallel increase of FAK and CAS phosphorylation. Knocking down Met or inhibition of endocytosis, but not Met pharmacological inhibition, reduced FAK and Cas phosphorylation. Knocking down Met or pharmacological FAK inhibition but not Met pharmacological inhibition inhibited cell migration.

The model proposed is that the two newly discovered NRP1 splice variant, through loss of N-glycosylation, co-internalise and recycle with Met, leading to increased stability of NRP1 splice variants and of Met and persistent FAK-Cas signalling on endosome and subsequent increase in cell migration and invasion.

This study is mostly strong and exciting, providing novel mechanisms of co-signalling of NRP1 and Met leading to enhanced migration and possibly metastasis. The fact that novel NRP1 splice variants enhance endosomal signalling in colorectal cancer may have future therapeutic significance. Overall, this is a nice novel story with mostly well performed experiments. The paper

is also well written and easy to follow.

There are three main points requiring clarification:

1- Some data are obtained in non-stimulated cells, but apparently in full serum (this is not very well explained but indicated in a couple of figure legends and in the discussion) while other experiments are obtained in cells stimulated with growth factors, apparently in serum deprived medium. Do the author assume the same mechanism occur in both conditions? As in presence of serum, NRP1 variants constitutively internalise while they internalise upon HGF stimulation in serum-starved conditions, do we assume that the basal internalisation is triggered by HGF present in the serum? This should be clarified.

2- How can we explain that Met pharmacological inhibition has no effect in any of the observed phenotypes? such on FAK-Cas signalling or cell migration while c-Met shRNA knock down does? Does it mean there is another kinase in the complex? This should be clarified. How are the authors sure that PHA-665752 (1 microM) has fully inhibited Met phosphorylation? Are other downstream signals than Cas and FAK reduced (such as ERK1/2 and AKT)?

3- How the defect in N-glycosylation in NRP1 trigger enhanced NRP1 and Met co-internalisation? Some mechanistic investigation would be warranted.

Specific comments

As HT2116 express the WT, DE4 and DE5 NRP1 mRNAs (Fig 1d), it is assumed they express the 3 protein forms. Thus why these cells were used for transfection of the NRP1 vectors?

Figure 1D should be supplemented by a Western blot to show protein expression.

Figure S2D: the intracellular localisation of endogenous NRP11 mutants in the primary cell lines is not very convincing. In Pt93 and Pt2377 cells, the staining appear localised in the Golgi (precursor form?); In LM2377, it is not clear if it is intracellular. Co-staining with an endosomal marker would be required.

Fig. 6a and b: IgG controls are required to control for NRP1 IP. Quantitative data on n=3 experiments are also required to clearly show the increased association between Met and splice variants compared to WT NRP1. Colocalization of WT NRP1 with Met at the plasma membrane is clearly visible. It is possible that that association is not increased overall.

Figs. 7e and 8 f: results with siCtr are missing.

NCOMMS-18-30740 Response to Reviewers

We sincerely thank the reviewers for carefully reading through our manuscript and providing valuable and constructive comments that have dramatically helped us to improve our manuscript and the presentation of our work. We have taken the comments from all reviewers seriously and revised our manuscript extensively. We believe that with our new data, the reversion has dramatically strengthened our study and addressed the reviewers' concerns. Below, we respond to the comments made by each reviewer.

Reviewer #1 (Remarks to the Author):

This is an interesting manuscript, suggesting the potential role of two specific glycosylation sites of NRP1 in c-Met driven metastatic progression of colorectal cancer. The experiments are well designed and mostly in agreement with the interpretation of the authors. This reviewer has the following comments.

1) What percentage of colorectal cancer tissues has these specific exon 4 and 5 deletions of NRP1? Do exon 4 and 5 deleted NRP1 (especially when consider along with cMET) correlate with colorectal cancer progression.

Response: We used our established RT-PCR method as described in the initial manuscript to further analyze the mRNA expression of the two NRP1 splice variants, NRP1- Δ E4 and NRP1- Δ E5, from a larger number of patients (n=126) with stage I-IV colorectal cancer (CRC). We found that NRP1- Δ E4 was positively expressed in 78% CRC tissues and significantly enriched in high-stage (III and IV) CRC tissues (63%) and associated with CRC progression, whereas NRP1- Δ E5 was expressed less (30%) in CRC tissues and not significantly observed as tumors progressed through stage I-IV. These new data are included as Fig. 1g,h and Supplementary Table 2, and described on pages 5-6 of the revision.

2) Fig 2: For Fig B and C, immunohistochemistry should be repeated with a known cell membrane protein. Z-stack image should be provided to confirm intracellular localization. It seems that exon 4 of NRP1 has been deleted in both the Pt2377 and LM2377. No result for endogenous deletion of exon 5 in colorectal cancer tissue has been provided, although the authors have mentioned such result in the text of the manuscript.

Response: We appreciate the constructive comments from Reviewer 1. As suggested, we used the membrane protein α 6-integrin as a positive control and repeat the immunofluorescence staining for NRP1-WT, NRP1- Δ E4 and NRP1- Δ E5 as shown in Fig. 2b, c in the original manuscript. The new data with Z-stack images confirmed that NRP1-WT was expressed at the plasma membrane, whereas NRP1- Δ E4 and NRP1- Δ E5 were localized predominately in intracellular compartments (Fig. 2b, c). Using the recombinant NRP1- Δ E4 and NRP1- Δ E5 proteins expressing in HCT116 cells as positive controls, we were able to detect the expression of both endogenous NRP1- Δ E4 and NRP1- Δ E5 in Pt93 primary cells (Supplementary Fig. 2e in the revision). While the molecular weights of NRP1- Δ E4 and NRP1- Δ E5 are quite close, NRP1- Δ E5 protein band could be separated and observed at the upper NRP1- Δ E4 protein band in Pt93 primary cells (Supplementary Fig. 2e in the revision). The text has been revised to specifically indicate the expression of both endogenous NRP1- Δ E4 and NRP1- Δ E5 in Pt93 primary cells on page 6.

3) Quantitation of supplement figure S3b and 3e should be provided. This experiment should be repeated with other lysosomal markers. This is an important information, which would provide that these splice variants are not delivered to lysosomes for degradation.

Response: As suggested by Reviewer 1, we provided the quantitation data for the co-localization of NRP1- Δ E4 or NRP1- Δ E5 with lysotracker in HT29 and HCT116 cells (Supplementary Fig. 3c, g in the revision). Additionally, we have repeated the experiments using lysosomal cathepsin D protease as an additional lysosomal marker. Consistent with our other data, we find that the two NRP1 splice variants were barely detected in lysosomes (Supplementary Fig. 3b, c, f, g in the revision), indicating that the two NRP1 variants do not significantly traffic to lysosomes for degradation. These new data are described on page 7.

4) *Figure 6C is interesting as it is suggesting that the internalization of c-Met is entirely dependent of NRP1. I am not sure whether this experiment has been performed in the presence of HGF. Do the authors mean that without NRP1 there will be no ligand-induced internalization of c-Met.*

Response: We believe that Met can internalize upon HGF stimulation by adding exogenous HGF to the serum-starved cells as we showed in Supplementary Figs. 6f and 10b in the revision. However, under the basal condition in regular cell growth medium containing 10% FBS with low levels of HGF (Supplementary Fig. 2h in the revision), Met could not be internalized and localized on the plasma membrane in WT NRP1-expressing cells (Fig. 6c in the revision). By contrast, both NRP1- Δ E4 and NRP1- Δ E5 could internalize under the basal condition (Fig. 2b,c). However, this internalization was dramatically reduced by depletion of HGF from FBS using an anti-HGF neutralizing antibody (Supplementary Fig. 2g,h in the revision). These data indicate that the two NRP1 variants are more sensitive to HGF for induction of internalization than the Met receptor. Furthermore, we found that both NRP1- Δ E4 and NRP1- Δ E5 had significantly enhanced interaction with Met compared to NRP1-WT in the basal condition (Fig. 6a,b in the revision). In such condition with low levels of HGF, Met co-internalized and co-localized with the two NRP1 variants on endosomes, but Met could not internalize in cells expressing NRP1-WT (Fig. 6c in the revision). In addition, depletion of either NRP1- Δ E4 or NRP1- Δ E5 expression completely prevented Met internalization in the basal condition (Fig. 6f in the revision). Collectively, these data strongly indicate that under basal conditions, low level of HGF is not sufficient to induce Met internalization, but is sufficient to induce the internalization of the two NRP1 splice variants, which result in Met co-internalization and co-localization on endosomes. These data provide key pieces of support for our dynamic trafficking model (Fig. 9 in the revision). The text has been revised to further describe these results regarding the differences between the internalization of the NRP1 variants, NRP1-WT and Met under basal conditions and upon HGF stimulation on pages 7 and 11-13.

5) *Endosomal C-Met mediated Rac1 signaling has been reported for enhanced cell migration. Sustained Rac1 activation therefore should be the read out for the endosomal cMet signaling and cell migration.*

Response: We agree with the point raised by Reviewer 1. We performed the Rac1 activity assay and found that Rac1 was substantially activated in HCT116 cells expressing either of the NRP1 splice variants in the regular cell growth medium containing 10% FBS or upon HGF stimulation (Supplementary Fig. 8a,b). In addition, silencing Rac1 profoundly inhibited cell migration in these cells (Supplementary Fig. 8c,d). We have included these new data in the revised manuscript as Supplementary Fig. 8, and described the results on page 13.

Reviewer #2 (Remarks to the Author):

This study from Huang and co-authors investigates the function of two isoforms of the transmembrane receptor Neuropilin-1 (NRP1) generated by exon skipping in colorectal cancer cells. In these isoforms, N-glycosylation is defective and causes aberrant trafficking of the NRP1 variants that are internalized and constitutively recycled to the cell surface, escaping the degradative pathway.

NRP1 is well known to interact with Receptor Tyrosine Kinases (RTKs), including MET, and regulate endocytosis of receptors on the cell surface. The Authors found here that an increased association of these novel NRP1 mutants with MET promotes their concomitant internalization, resulting in MET accumulation in endosomes. This is associated with persistent activation of the FAK/p130Cas signaling pathway, ultimately promoting invasion and metastatic dissemination of colon cancer cells.

The quality of the data presented here is good and the statistical analysis meaningful, even if a non-parametric tool such as the Mann-Whitney test would have been more appropriate than a two-tailed Student t-test. Investigation of receptor trafficking has been performed using a wealth of tools and approaches generating convincing results. Methods are adequately described.

However, the preliminary evidence that inhibition of MET kinase phosphorylation by PHA-665752 does not prevent FAK activation induced by NRP1 variants, nor the downstream effects attributed to MET function, poses the question about the actual implicated mechanisms, which are not elucidated here.

Major points:

1) According to the proposed working model, NRP1 variants promote MET endocytosis in HGF-stimulated cells. However, NRP1 isoforms and MET are also internalized under basal conditions in the presence of FBS (Figure 2b,c and Figure 6f), but not in starved cells (Figure 2f). Is this also accounted by HGF-dependent MET activation? or by the involvement of other factors? Since HGF-induced MET internalization is known to depend on its tyrosine kinase activity, one would expect that NRP1 variants-associated endocytosis were also dependent on MET kinase activity, while this was surprisingly excluded by data shown in Suppl. Fig. 7b. How can this apparent conflict be explained?

Response: This is an insightful comment from Reviewer 2. As noted in a previous response to Reviewer 1's comment 4, we showed that under basal conditions in the presence of FBS, low level of HGF is not sufficient to induce internalization of Met or NRP1-WT, but is sufficient to induce the co-internalization of NRP1 splice variants (NRP1- Δ E4, NRP1- Δ E5) and Met (Fig. 6c; Supplementary Fig. 2g,h). Silencing Met did not block the internalization of NRP1- Δ E4 and NRP1- Δ E5 under the basal condition, but silencing either NRP1- Δ E4 or NRP1- Δ E5 completely prevented the internalization of Met (Fig. 6e,f). In the presence of HGF stimulation, the Met inhibitor, PHA-665752, did block Met internalization in HCT116 cells (Supplementary Fig. 10b), which is consistent with the view that HGF-induced MET internalization depends on its tyrosine kinase activity¹. However, the Met internalization was not blocked by PHA-665752 in HCT116 cells expressing NRP1- Δ E4 or NRP1- Δ E5 in the regular growth medium (Supplementary Fig. 10c). In the revised manuscript, we further found that the two NRP1 variants also recruited β 1-integrin to form the NRP1 variants/Met/ β 1-integrin complexes, and these complexes co-internalized into endosomes under basal conditions (Fig. 6a-g). Several studies have highlighted the functional importance of the cross-activating Met/ β 1-integrin complex in a ligand-independent manner to drive cancer invasion and metastasis²⁻⁴. In addition, GIPC1 is a well-known NRP1 interacting protein via direct binding to the SEA motif in the intracellular domain of NRP1⁵, and couples NRP1 to other signaling receptors, including VEGFR and integrins, to accelerate their endocytosis^{6,7}. Interestingly, silencing GIPC1 by siRNA largely abrogated the internalization of the two NRP1 variants as well as the co-internalization of Met and β 1 integrin in the NRP1 variant-expressing cells under basal conditions (Fig. 6h,i). While the regulation of the endocytosis of the NRP1 variants/Met/ β 1-integrin complexes need to be further characterized in more detail, our accumulated findings in the revised manuscript strongly indicate that the physical interaction

with the NRP1 variants is required for Met/ β 1-integrin co-internalization, which is mediated by GIPC1, but does not depend on Met kinase activity under basal conditions.

We have included these new data in the revised manuscript as Fig. 6a-i and Supplementary Figs. 2g,h, 10b,c, described the results on pages 7, 11-13 and 15; and discussed the data in detail on page 20.

2) The functional impact of NRP1 variants in cancer cell migration and invasion is shown to be dependent on MET and FAK expression, by means of RNAi-based experiments (in Fig. 6i-j and 8f). However, the relevance of this pathway associated with the expression of NRP1 variants should be validated also for the induction of metastatic spreading in vivo, which is particularly relevant in translational perspective.

Response: We agree with the point raised by Reviewer 2 and have now performed the experiments as suggested. Using an experimental lung metastasis model *in vivo*, we demonstrated that shRNA-mediated knockdown of Met or FAK expression dramatically repressed the formation of lung metastasis *in vivo* induced by NRP1- Δ E4 (Figs. 6l,m and 8j-l). As suggested by Reviewer 2, we used the Mann-Whitney test to perform the statistical analysis of these animal studies. We have included these new data in the revised manuscript as Figs. 6l,m and 8j-l, and described the results on pages 13, 16, 42 and 44.

3) The presented experiments are almost always based on overexpression of the NRP1 variants in cancer cells. It is understandable the difficulty to target specifically the endogenous splice variants without affecting the wild type transcript. However, some data should be provided to validate the relevance of endogenously expressed NRP1 variants in cancer cells. For instance, the Authors could assay their preferential co-immunoprecipitation with MET, as claimed based on data shown in Figure 6a. In fact, Fig. S2c shows that the endogenous variants can be detected and clearly discriminated from WT-NRP1 by western blotting.

Response: As suggested by Reviewer 2, we performed the co-immunoprecipitation assay and found that the endogenous NRP1- Δ E4 and/or NRP1- Δ E5 that are dominantly expressed in the primary CRC cell lines (Pt93, Pt2377 and LM2377) increased their binding to Met than the endogenous NRP1-WT expressed only in the Pt130 primary CRC cell line (Supplementary Figs. 2e and 6c). These data are similar to that obtained with exogenous expression of NRP1- Δ E4 or NRP1- Δ E5 as compared with NRP1-WT expression in HT29 cells (Fig. 6a). We have included these new data in the revised manuscript as Supplementary Fig. 6c, and described the results on page 12.

4) NRP1 variants-induced MET internalization is proposed to activate FAK signaling, and MET downregulation by RNAi can block the pathway. However, preliminary data shown in Supplementary Figure S7a seem to suggest that FAK activation here is not due to MET kinase activity, as it is not blocked by PHA-665752 inhibitor. Actually, direct interaction between MET and FAK has been reported previously, and indeed it depends on MET phosphorylation (DOI:10.1128/MCB.02186-05). This is in contrast with findings presented here and challenges the relevance of data shown in Figure S7, as no alternative signaling mechanism is proposed to account for FAK regulation by MET in this study. In general, it is puzzling that the treatment with an inhibitor of MET kinase activity, which is known to be crucial for HGF-induced regulation of cell migration and invasion, is unable to block the downstream effects induced by NRP1 variants and mediated by MET expression. At the present stage, I would recommend to reconsider the relevance to these data for the study. One option could be to leave out Figure S7, acknowledging that the mechanism of FAK activation by MET in cells overexpressing NRP1 variants needs to be further investigated. In any case, the involvement of MET in this pathway independent of its kinase activity should be accurately discussed, in face of the current literature.

Response: We appreciate the insightful comments from Reviewer 2. While Met has been shown to directly binds to FAK leading to FAK activation in a Met phosphorylation-dependent manner⁸, the active β 1-integrin can also positively regulate FAK activity on endosomes⁹. Using the co-immunoprecipitation assay and the confocal microscopy analysis, we further identified β 1-integrin as an additional binding receptor with the two NRP1 variants and Met to form the NRP1 variants/Met/ β 1-integrin complexes, and these complexes co-internalized to endosomes under basal conditions (Fig. 6a-g and Supplementary Fig. 7). Notably, silencing β 1-integrin profoundly inhibited phosphorylation of FAK and CRC cell migration, invasion and metastasis induced by the NRP1 variants (Figs. 6j-m and 8e), although β 1-integrin knockdown did not affect the co-localization of the NRP1 variants and Met on endosomes (Fig. 6g). Furthermore, knockdown of Met expression caused β 1-integrin dissociation from the endosomal NRP1 variants with concomitant inhibition of FAK phosphorylation (Figs. 6f and 8b). However, the endocytosis of the NRP1 variants/Met/ β 1-integrin complexes on endosomes associated with activation of FAK and cell migration/invasion was not inhibited by the Met inhibitor PHA-665752 (Supplementary Fig. 10a, c-e). Taken together, our new findings suggest that the endosomal β 1-integrin bypass signaling contributes to FAK activation and the Met tyrosine kinase inhibitor-resistance in CRC cells expressing the NRP1 variants. These findings also highlight the functional importance of the NRP1 variants/Met/ β 1-integrin complexes on endosomes in activating FAK signaling for CRC cell dissemination. We have included these new data in the revised manuscript as Figs. 6a-g, j-m and 8e and Supplementary Figs. 7 and 10a, c-e, described the results on pages 11-13; and discussed the data in detail on page 20.

5) *Notably, NRP1 has been shown also to interact with Integrins (doi: 10.1038/s41388-018-0290-4), which are major regulators of FAK activation in endosomes (doi: 10.1038/s41388-018-0290-4, doi: 10.1038/ncb3250, doi: 10.1016/j.tcb.2016.02.001). Therefore, the possibility that, along with MET, integrins are constitutively internalized by NRP1 variants and activate FAK on endosomes should be addressed.*

Response: See below.

6) *Endosomal activation of FAK has been demonstrated in several studies. The authors suggest that, in the colon cancer cell lines analyzed here, FAK activation is due to persistent MET signaling from endosomes, which, in turn, is driven by NRP1 variants. However, visualization of FAK activation on endosomes positive for NRP1 isoforms would strengthen the proposed model.*

Response: We greatly appreciate the constructive and valuable comments from Reviewer 2. Since comment 6 is related to comment 5, we would like to address these two issues jointly.

As noted in the response to Reviewer 2's comment 4 above, we identified β 1-integrin as a key and additional binding receptor with the two NRP1 variants and Met to form the NRP1 variants/Met/ β 1-integrin complexes, and these complexes co-internalized and accumulated on endosomes (Fig. 6a-g and Supplementary Fig. 7). Silencing β 1-integrin did not affect the internalization of the NRP1 variants under basal conditions, but silencing NRP1 variants prevented the internalization of active β 1-integrin (Fig. 6g). Similar to the Met co-internalization with the NRP1 variants as noted in the response to Reviewer 2's comment 1, the NRP1 variants also regulated the internalization of active β 1-integrin mediated by GIPC1 and they co-localized on endosomes (Fig. 6g-i). Consistent with the reported activation of FAK by endosomal β 1-integrin⁹, we further found that active β 1-integrin, phosphorylated FAK (active form) and the NRP1 variants were all co-localized on endosomes (Fig. 8c,d), and silencing β 1-integrin profoundly inhibited phosphorylation of FAK and CRC cell migration, invasion and metastasis induced by the NRP1 variants (Figs. 6j-m and 8e). Collectively, these findings highlight the functional importance of β 1-integrin for the endosomal activation of FAK induced by the NRP1 variants.

We have included these new data in the revised manuscript as Figs. 6, 8c-e and Supplementary Fig. 7, described the results on pages 11-13 and 15; and discussed the data in detail on page 20.

Additional points:

7) *Semi-quantitative analysis of NRP1 variants expression shown in Fig. 1e should be validated by comparison to the levels of a housekeeper gene mRNA in the same samples. It is surprising that normal colonic mucosa does not express any NRP1 transcript.*

Response: As suggested by Reviewer 2, we provided the quantitation data for the expression levels of NRP1 variants by comparison with GAPDH mRNA expression levels in the same samples (Fig. 1f). In the original manuscript, no NRP1 transcripts were detected in normal colonic mucosa as shown in Fig. 1e is because of using lower amount of RNA for RT-PCR analysis. Indeed, WT *NRP1* could be detected in these normal tissues by increasing their RNA levels (Supplementary Fig. 2a). These new data are now included in the revised manuscript, and described on page 5.

8) *Concerning results shown in Figure 2, a western blot revealing VEGFR, EGFR and MET protein levels in the cell lines under analysis should be provided, in order to assess the relative expression of these receptors.*

Response: We performed the experiments as suggested by Reviewer 2. We found that expression of NRP1-WT, NRP1- Δ E4 or NRP1- Δ E5 in HCT116 and HT29 cells did not affect the protein levels of EGFR and Met receptors (Fig. 2a), whereas VEGFR2 expression was not detected in these two cell lines using western blot analysis (data not shown). We have included these new data in the revised manuscript as Fig. 2a, and described the results on page 6.

9) *With reference to data shown in Fig. 3d-e, in midpage 7, it is said “After 15 min incubation with culture medium”; it should be specified what kind of medium was used here? FBS-containing or serum-free?*

Response: We have revised the text accordingly to “After 15 min incubation with culture medium containing 10% FBS” on page 8.

10) *The relevance of data shown in Supplementary Fig. 5f-g is questionable. In fact, how can the impact of WT vs. splice variant NRP1 be discriminated here? Indeed the cells express all mRNA isoforms. This major limitation should at least be highlighted in the manuscript.*

Response: We agree with the point raised by Reviewer 2 and have now revised the text accordingly to “silencing NRP1 in both Pt93 and LM2377 primary CRC cells that express higher levels of the endogenous NRP1- Δ E4 and/or NRP1- Δ E5 than NRP1-WT (Supplementary Fig. 2e) resulted in a marked inhibition of cell migration and invasion (Supplementary Fig. 5e-g).” on page 10 for clarification.

11) *Data plotting in Fig. 8f is not appropriate, as the Y-axis scale is severely cut to values above 60%, and the basal impact of FAK silencing on migration is not reported. Absolute scores of cell migration should be plotted, or % inhibition of controls by showing the entire range 0-100%.*

Response: We appreciate this constructive comment from Reviewer 2. As suggested, we include the results with control siRNA (siCtrl) and presented the data as the fold change over the migrated cell number found in NRP1-WT-expressing cells transfected with siCtrl in Fig. 8i (Fig. 8f in the original manuscript) in

the revised manuscript.

12) *In the study, it is also investigated the potential regulatory role of NRP1 variants on EGFR tyrosine kinase. This is indeed justified by previous literature linking wild type NRP1 to EGFR in other experimental models (e.g. DOI: 10.1158/0008-5472.CAN-12-0995). Even if the results shown here in colon cancer cells do not confirm this putative working hypothesis, in order to put the data into context, the relevant previous literature should be cited.*

Response: We agree with the point raised by Reviewer 2, and have included the reference in the text on page 3.

13) *The text needs to be carefully edited for typos. For instance, in page 4 line 8: "Furthermore, our work highlight that..."; or in page 7 line 9: "may secret to the extracellular space..."; etc...*

Response: We appreciate for making these corrections from Reviewer 2. We have revised as requested.

Reviewer #3 (Remarks to the Author):

This manuscript from Qing-Bai She and colleagues reports the discovery of neuropilin-1 (NRP1) splice variants in colorectal cancer. These two NRP1 variants are shown to constitutively internalise, recycle and to be protected from degradation. It is further shown that these mutants miss a distinct glycosylation site each. Mutants of either of these glycosylation sites phenocopy the splice variant mutants for the constitutive internalisation and increased stability. Splice variants and glycosylation mutants enhance cell migration, invasion and in vivo lung colonisation. Co-immunoprecipitation and colocalization in vesicles between Met receptor and the NRP1 splice variants are observed. Moreover, Met internalisation and stability are increased in cells expressing NRP1 splice variants versus cells expressing NRP1 WT. It is further shown that blocking endocytosis with dynasore, clathrin knock down or expression of dynamin dominant negative inhibit NRP1 variants and Met co-internalisation and cell migration and invasion. Met was found phosphorylated in the cells expressing NRP1 variants with a parallel increase of FAK and CAS phosphorylation. Knocking down Met or inhibition of endocytosis, but not Met pharmacological inhibition, reduced FAK and Cas phosphorylation. Knocking down Met or pharmacological FAK inhibition but not Met pharmacological inhibition inhibited cell migration.

The model proposed is that the two newly discovered NRP1 splice variant, through loss of N-glycosylation, co-internalise and recycle with Met, leading to increased stability of NRP1 splice variants and of Met and persistent FAK-Cas signalling on endosome and subsequent increase in cell migration and invasion.

This study is mostly strong and exciting, providing novel mechanisms of co-signalling of NRP1 and Met leading to enhanced migration and possibly metastasis. The fact that novel NRP1 splice variants enhance endosomal signalling in colorectal cancer may have future therapeutic significance. Overall, this is a nice novel story with mostly well performed experiments. The paper is also well written and easy to follow.

There are three main points requiring clarification:

1) *Some data are obtained in non-stimulated cells, but apparently in full serum (this is not very well explained but indicated in a couple of figure legends and in the discussion) while other experiments are obtained in cells stimulated with growth factors, apparently in serum deprived medium. Do the author assume the same mechanism occur in both conditions? As in presence of serum, NRP1 variants*

constitutively internalise while they internalise upon HGF stimulation in serum-starved conditions, do we assume that the basal internalisation is triggered by HGF present in the serum? This should be clarified.

Response: We greatly appreciate the insightful comments from Reviewer 3. We agree with Reviewer 3's assumption that the constitutive internalization of the NRP1 variants in presence of serum is triggered by HGF present in the serum. To address this concern, we used an anti-HGF neutralizing antibody to deplete HGF from the serum. As compared to the constitutive internalization of the NRP1 variants induced by 10% FBS in the cell growth medium, the HGF-neutralized FBS was unable to stimulate significant internalization of the NRP1 variants (Supplementary Fig. 2g,h). Thus, our findings indicate that HGF is a critical ligand to induce the internalization of the NRP1 splice variants. We have included these new data in the revised manuscript as Supplementary Fig. 2g,h, and described the results on page 7. We also revised the text accordingly to indicate the difference between the basal condition and serum-starved condition upon HGF stimulation for clarification.

2) How can we explain that Met pharmacological inhibition has no effect in any of the observed phenotypes? such on FAK-Cas signalling or cell migration while c-Met shRNA knock down does? Does it mean there is another kinase in the complex? This should be clarified. How are the authors sure that PHA-665752 (1 microM) has fully inhibited Met phosphorylation? Are other downstream signals than Cas and FAK reduced (such as ERK1/2 and AKT)?

Response: We appreciate this illuminating comment from Reviewer 3. This comment is similar to the request by Reviewer 2. As described in our responses to Reviewer 2's comments 4-6, we identified β 1-integrin as a key and additional binding receptor with the two NRP1 variants and Met to form the NRP1 variants/Met/ β 1-integrin complexes, and these complexes co-internalized and accumulated on endosomes to substantially activate FAK/p130Cas signaling pathway for CRC cell migration, invasion and dissemination (Fig. 6 and Supplementary Fig. 7). Consistent with the reported activation of FAK by endosomal β 1 integrin⁹, we reiterate that active β 1-integrin, Met, phosphorylated FAK (active form) and the NRP1 variants were all co-localized on endosomes (Figs. 6f,g and 8c,d), and silencing β 1-integrin profoundly inhibited phosphorylation of FAK and CRC cell migration, invasion and metastasis induced by the NRP1 variants (Figs. 6j-m and 8e), although β 1-integrin knockdown did not affect Met interaction and co-localization with the NRP1 variants on endosomes (Fig. 6g and Supplementary Fig. 7g). Interestingly, knockdown of Met expression caused β 1-integrin dissociation from the endosomal NRP1 variants with concomitant inhibition of FAK/p130Cas signaling and CRC cell migration, invasion and dissemination (Figs. 6f,j-m and 8b). However, activation of FAK and cell migration/invasion induced by the NRP1 variants were not inhibited by the Met inhibitor PHA-665752 in HCT116 cells (Supplementary Fig. 10 a, c-e). In this study, we used 1 μ M PHA-665752, a concentration previously shown to potently inhibit Met activity in multiple Met-dependent and -independent cancer cell lines¹⁰⁻¹³. As suggested by Reviewer 3, we further tested the phosphorylation levels of AKT and ERK, and also found no inhibitory effect on AKT and ERK activation by PHA-665752 in HCT116 cells expressing either NRP1-WT or the NRP1 variants (Supplementary Fig. 10a). This Met independence is consistent with a recent study showing that NRP1 upregulation elicits adaptive resistance to oncogene-targeted therapies including the use of Met inhibitors¹⁴. In addition, KRAS and PI3K mutations leading to constitutive activation of AKT and ERK signaling are frequently observed in CRC (e.g., HCT116 cells)¹⁵⁻¹⁸, and can cause CRC resistance to upstream tyrosine kinase receptor-targeted therapies^{19,20}. Furthermore, several studies have highlighted an important role of the cross-activating Met/ β 1-integrin complex in a ligand-independent manner to drive cancer invasion and metastasis²⁻⁴. Collectively, our new findings strongly indicate that the endosomal β 1-integrin bypass signaling contributes to FAK activation and renders Met tyrosine kinase inhibitor-resistance in CRC cells expressing the NRP1 variants. Thus, our findings highlight the functional importance of the

NRP1 variants/Met/ β 1-integrin complexes on endosomes in activating FAK signaling for CRC metastasis.

We have included these new data in the revised manuscript as Figs. 6 and 8b-e and Supplementary Figs. 7 and 10a-e, described the results on pages 11-13; and discussed the data in detail on page 20.

3) How the defect in N-glycosylation in NRP1 trigger enhanced NRP1 and Met co-internalisation? Some mechanistic investigation would be warranted.

Response: As noted in a previous response to Reviewer 2's comment 1, we demonstrated that silencing GIPC1, a well-known NRP1 interacting protein functioned as an endocytic adaptor⁵⁻⁷, attenuated the internalization of the NRP1 variants under basal conditions. In addition, GIPC1 knockdown also largely abrogated the co-internalization of Met and β 1 integrin with the NRP1 variants. While the regulation of the endocytosis of the NRP1 variants/Met/ β 1 integrin complexes need to be further characterized in more detail, our accumulated findings in the revised manuscript strongly suggest that GIPC1 is key endocytic adaptor protein that mediates the co-internalization of Met/ β 1 integrin with the NRP1 variants.

We have included these new data in the revised manuscript as Fig. 6h,i, described the results on page 13; and discussed the data in detail on page 20.

Specific comments

4) As HCT116 express the WT, DE4 and DE5 NRP1 mRNAs (Fig 1d), it is assumed they express the 3 protein forms. Thus why these cells were used for transfection of the NRP1 vectors?

Response: While the mRNA expression of NRP1-WT, NRP1- Δ E4 and NRP1- Δ E5 could be detected in HCT116 cells by RT-PCR, their protein expression levels were barely detected using western blot analysis (Supplementary Fig. 2b in the revision). The text has been revised for clarification on page 6.

5) Figure 1D should be supplemented by a Western blot to show protein expression.

Response: As suggested by Reviewer 3, we performed the western blot analysis of NRP1 protein expression in HCT116, HT29 and DLD-1 cells. These new data are now included in the revised manuscript as Supplementary Fig. 2b, and described on page 6.

6) Figure S2D: the intracellular localisation of endogenous NRP1 mutants in the primary cell lines is not very convincing. In Pt93 and Pt2377 cells, the staining appear localised in the Golgi (precursor form?); In LM2377, it is not clear if it is intracellular. Co-staining with an endosomal marker would be required.

Response: We agree with the points raised by Reviewer 3 and have now performed the experiments as suggested. Our results showed that the intracellular accumulation of endogenous NRP1- Δ E4 and NRP1- Δ E5 in the primary CRC cell lines (Pt93, Pt2377 and LM2377) was also found in endosomes as demonstrated by co-localization with the endosomal marker Rab7. These new data are now included in the revised manuscript as Supplementary Fig. 3h, and described on pages 7-8.

7) Fig. 6a and b: IgG controls are required to control for NPR1 IP. Quantitative data on n=3 experiments are also required to clearly show the increased association between Met and splice variants compared to WT NRP1. Colocalization of WT NRP1 with Met at the plasma membrane is clearly visible. It is possible that that association is not increased overall.

Response: We agree with Reviewer 3 that IgG controls are required to control for NPR1 IP. In the original

manuscript, we did use the IgG as a NRP1 IP control. We apologize for not presenting IgG as IP control clearly, which may have caused Reviewer 3 confusion. We have now presented the IgG control for IP more clearly in Fig. 6a of the revision. As suggested by Reviewer 3, we have now provided the quantitative data from three independent IP experiments showing that the two NRP1 splice variants had greater interaction with Met and with $\beta 1$ integrin as well compared with WT NRP1 (Fig. 6b). In addition, the percentages of the NRP1 variants co-localization with Met or $\beta 1$ integrin in intracellular compartments were also significantly higher than those of WT NRP1 co-localization with Met or $\beta 1$ integrin on the plasma membrane as analyzed by using Nikon NIS-Elements AR software (Fig. 6c,d). We have included these new data in the revised manuscript as Fig. 6a-d and described the results on page 11.

8) *Figs. 7e and 8 f: results with siCtrl are missing.*

Response: As suggested by Reviewer 3, we have now included the results with siCtrl in Figs. 7e and 8i (Fig. 8f in the original manuscript) in the revised manuscript.

References

- 1 Li, N., Lorinczi, M., Ireton, K. & Elferink, L. A. Specific Grb2-mediated interactions regulate clathrin-dependent endocytosis of the cMet-tyrosine kinase. *J. Biol. Chem.* **282**, 16764-16775 (2007).
- 2 Barrow-McGee, R. *et al.* Beta 1-integrin-c-Met cooperation reveals an inside-in survival signalling on autophagy-related endomembranes. *Nat. Commun.* **7**, 11942 (2016).
- 3 Jahangiri, A. *et al.* Cross-activating c-Met/beta1 integrin complex drives metastasis and invasive resistance in cancer. *Proc. Natl. Acad. Sci. USA* **114**, E8685-E8694 (2017).
- 4 Mitra, A. K. *et al.* Ligand-independent activation of c-Met by fibronectin and alpha(5)beta(1)-integrin regulates ovarian cancer invasion and metastasis. *Oncogene* **30**, 1566-1576 (2011).
- 5 Cai, H. & Reed, R. R. Cloning and characterization of neuropilin-1-interacting protein: a PSD-95/Dlg/ZO-1 domain-containing protein that interacts with the cytoplasmic domain of neuropilin-1. *J. Neurosci.* **19**, 6519-6527 (1999).
- 6 Wang, L., Mukhopadhyay, D. & Xu, X. C terminus of RGS-GAIP-interacting protein conveys neuropilin-1-mediated signaling during angiogenesis. *FASEB J.* **20**, 1513-1515 (2006).
- 7 Valdembri, D. *et al.* Neuropilin-1/GIPC1 signaling regulates alpha5beta1 integrin traffic and function in endothelial cells. *PLoS Biol.* **7**, e25 (2009).
- 8 Chen, S. Y. & Chen, H. C. Direct interaction of focal adhesion kinase (FAK) with Met is required for FAK to promote hepatocyte growth factor-induced cell invasion. *Mol. Cell Biol.* **26**, 5155-5167 (2006).
- 9 Alanko, J. *et al.* Integrin endosomal signalling suppresses anoikis. *Nat. Cell Biol.* **17**, 1412-1421 (2015).
- 10 Smolen, G. A. *et al.* Amplification of MET may identify a subset of cancers with extreme sensitivity to the selective tyrosine kinase inhibitor PHA-665752. *Proc. Natl. Acad. Sci. USA* **103**, 2316-2321 (2006).
- 11 Engelman, J. A. *et al.* MET amplification leads to gefitinib resistance in lung cancer by activating ERBB3 signaling. *Science* **316**, 1039-1043 (2007).
- 12 Turke, A. B. *et al.* Preexistence and clonal selection of MET amplification in EGFR mutant NSCLC. *Cancer cell* **17**, 77-88 (2010).
- 13 Qi, J. *et al.* Multiple mutations and bypass mechanisms can contribute to development of acquired resistance to MET inhibitors. *Cancer Res.* **71**, 1081-1091 (2011).
- 14 Rizzolio, S. *et al.* Neuropilin-1 upregulation elicits adaptive resistance to oncogene-targeted therapies. *J. Clin. Invest.* **128**, 3976-3990 (2018).
- 15 Parsons, D. W. *et al.* Colorectal cancer: mutations in a signalling pathway. *Nature* **436**, 792 (2005).
- 16 Cancer Genome Atlas, N. Comprehensive molecular characterization of human colon and rectal cancer. *Nature* **487**, 330-337 (2012).

- 17 She, Q. B. *et al.* 4E-BP1 is a key effector of the oncogenic activation of the AKT and ERK signaling pathways that integrates their function in tumors. *Cancer Cell* **18**, 39-51 (2010).
- 18 Ye, Q., Cai, W., Zheng, Y., Evers, B. M. & She, Q. B. ERK and AKT signaling cooperate to translationally regulate survivin expression for metastatic progression of colorectal cancer. *Oncogene* **33** (2014).
- 19 Zhao, B. *et al.* Mechanisms of resistance to anti-EGFR therapy in colorectal cancer. *Oncotarget* **8**, 3980-4000 (2017).
- 20 Xu, J. M. *et al.* PIK3CA Mutations Contribute to Acquired Cetuximab Resistance in Patients with Metastatic Colorectal Cancer. *Clin. Cancer Res.* **23**, 4602-4616 (2017).

REVIEWERS' COMMENTS:

Reviewer #1 (Remarks to the Author):

The authors have revised the manuscript appropriately. This reviewer has no further concern.

Reviewer #2 (Remarks to the Author):

In this revised version of the manuscript, the Authors have addressed most of my concerns and have shown many additional data, relevant to the conclusions.

I recommend the revised manuscript is carefully revised for English language. In fact, at several instances I found words or statements that may prove confusing to the readers.

For example:

- In the Abstract, the last part of the following statement is not clear: "This increased endocytic trafficking is triggered by loss of N-glycosylation at the Asn150 or Asn261 site in the two respective NRP1 variants upon HGF stimulation". Obviously the impact of HGF is not on N-glycosylation as it may sound. Maybe: "This increased endocytic trafficking of the two NRP1 variants, upon HGF stimulation, is due to loss of N-glycosylation at the Asn150 or Asn261 site, respectively".
- Line 291: It seems advisable to use "while" instead of "but"
- Line 224: It seems advisable to use "However" instead of "In contrast".
- Line 258: Correct the typo "predominately"
- Line 260: The frase "In contrast" is confusing; if fact, both WT and NRP1 mutants colocalize with Met and active β 1-integrin , right? Although, what changes is where they colocalize in the cell. This should be explained more clearly.
- Line 279: Correct the frase: "the two NRP1 variants remained localization"; maybe the Authors meant "localized"?
- Line 290: Confusing statement: "...and additionally, the two NRP1 variants render Met and β 1-integrin co-internalization and accumulation on endosomes through GIPC1 and their enhanced binding abilities with Met and β 1-integrin". Please clarify (and also maybe split the statement).
- Line 312: Check: "or expressed with"; did the Authors mean "transfected"?
- Line 464: Check the frase: "renders Met tyrosine kinase inhibitor-resistance in these cells"; did the Authors mean "inhibitor-resistant"?

Finally, Figure 9 is very useful for visualizing the complex signaling mechanism elucidated in this study. However, since all data seem to indicate that the kinase activity of Met is not implicated in the pathway (not even in response to HGF-stimulation), I believe that representing Met in its phosphorylated state (with circled P attached to the tail) maybe confusing to the readers, suggesting a different idea. My recommendation is to remove that mark from the scheme.

Reviewer #3 (Remarks to the Author):

I am happy with this revised version. All my queries have been fulfilled.

NCOMMS-18-30740A Response to Reviewers

We sincerely thank the reviewers for carefully reading through our revised manuscript and providing constructive comments that have helped us to dramatically improve our manuscript and the presentation of our work. Reviewer 1 and Reviewer 3 have no more concerns. Below, we respond to the comments made by Reviewer 2.

Reviewer #1 (Remarks to the Author):

The authors have revised the manuscript appropriately. This reviewer has no further concerns.

Reviewer #2 (Remarks to the Author):

In this revised version of the manuscript, the Authors have addressed most of my concerns and have shown many additional data, relevant to the conclusions.

I recommend the revised manuscript is carefully revised for English language. In fact, at several instances I found words or statements that may prove confusing to the readers. For example:

1) In the Abstract, the last part of the following statement is not clear: “This increased endocytic trafficking is triggered by loss of N-glycosylation at the Asn150 or Asn261 site in the two respective NRP1 variants upon HGF stimulation”. Obviously the impact of HGF is not on N-glycosylation as it may sound. Maybe: “This increased endocytic trafficking of the two NRP1 variants, upon HGF stimulation, is due to loss of N-glycosylation at the Asn150 or Asn261 site, respectively”.

Response: We agree with the point raised by Reviewer 2 and have revised the abstract as suggested.

2) Line 211: It seems advisable to use “while” instead of “but”

Response: We have changed “but” to “while” as suggested.

3) Line 224: It seems advisable to use “However” instead of “In contrast”.

Response: We have changed the text to state “However” instead of “In contrast”.

4) Line 258: Correct the typo “predominately”

Response: Thank you for pointing out the misspelling. We have changed “predominately” to “predominantly”.

5) Line 260: The frase “In contrast” is confusing; if fact, both WT and NRP1 mutants colocalize with Met and active β 1-integrin , right? Although, what changes is where they colocalize in the cell. This should be explained more clearly.

Response: We agree and have revised the text on page 11 to read “However, active β 1-integrin co-localized with NRP1-WT and Met on the plasma membrane in NRP1-WT-expressing cells under basal conditions (Fig. 6c,d).”

6) Line 279: Correct the frase: “the two NRP1 variants remained localization”; maybe the Authors meant

“localized”?

Response: You are correct. We meant “localized.” To shorten the sentence and add clarity, we re-wrote it to read: “Notably, knockdown of either Met or β 1-integrin expression did not alter the accumulation of NRP1- Δ E4 or NRP1- Δ E5 on endosomes or protein stability 8 h after cycloheximide exposure (Fig. 6e-g; Supplementary Figs. 6l,m, 7e,f).”

7) Line 290: Confusing statement: “...and additionally, the two NRP1 variants render Met and β 1-integrin co-internalization and accumulation on endosomes through GIPC1 and their enhanced binding abilities with Met and β 1-integrin”. Please clarify (and also maybe split the statement).

Response: As suggested, we split the statement on page 12 to read: “these data indicate that internalization of NRP1- Δ E4 and NRP1- Δ E5 and their accumulation on endosomes are independent of Met or β 1-integrin activity. Conversely, internalization of Met and β 1-integrin and their accumulation on endosomes are controlled by the NRP1 variants and the adaptor protein GIPC1.”

8) Line 312: Check: “or expressed with”; did the Authors mean “transfected”?

Response: Thank you for catching this. We have changed “expressed” to “transfected”.

9) Line 464: Check the frase: “renders Met tyrosine kinase inhibitor-resistance in these cells”; did the Authors mean “inhibitor-resistant”?

Response: We have changed “inhibitor-resistance” to “inhibitor-resistant”.

10) Finally, Figure 9 is very useful for visualizing the complex signaling mechanism elucidated in this study. However, since all data seem to indicate that the kinase activity of Met is not implicated in the pathway (not even in response to HGF-stimulation), I believe that representing Met in its phosphorylated state (with circled P attached to the tail) maybe confusing to the readers, suggesting a different idea. My recommendation is to remove that mark from the scheme.

Response: We appreciate this constructive comment from Reviewer 2, and have revised the scheme in Fig. 9 accordingly.

In summary, we have addressed all the concerns above raised by Reviewer 2. In addition, we have also carefully revised the manuscript for English language using the “track changes” feature in Word.

Reviewer #3 (Remarks to the Author):

I am happy with this revised version. All my queries have been fulfilled.